# Gated Slot Attention for Efficient Linear-Time Sequence Modeling

**Yu Zhang**[1*]  **Songlin Yang**[2*]  **Ruijie Zhu**[3]  **Yue Zhang**[1]  **Leyang Cui**[4]
**Yiqiao Wang**[5]  **Bolun Wang**[5]  **Freda Shi**[6]  **Bailin Wang**[2]
**Wei Bi**[4]  **Peng Zhou**[5†]  **Guohong Fu**[1†]

[1]School of Computer Science and Technology, Soochow University, China
[2]Massachusetts Institute of Technology   [3]University of California, Santa Cruz
[4]Tencent AI Lab   [5]LuxiTech   [6]University of Waterloo
yzhang.cs@outlook.com   yangsl66@mit.edu
 https://github.com/sustcsonglin/flash-linear-attention
 https://huggingface.co/fla-hub

## Abstract

Linear attention Transformers and their gated variants, celebrated for enabling parallel training and efficient recurrent inference, still fall short in recall-intensive tasks compared to traditional Transformers and demand significant resources for training from scratch. This paper introduces Gated Slot Attention (GSA), which enhances Attention with Bounded-memory-Control (ABC [63]) by incorporating a gating mechanism inspired by Gated Linear Attention (GLA [96]). Essentially, GSA comprises a two-layer GLA linked via $\mathrm{softmax}$, utilizing context-aware memory reading and adaptive forgetting to improve memory capacity while maintaining compact recurrent state size. This design greatly enhances both training and inference efficiency through GLA's hardware-efficient training algorithm and reduced state size. Additionally, retaining the $\mathrm{softmax}$ operation is particularly beneficial in "finetuning pretrained Transformers to RNNs" (T2R [41]) settings, reducing the need for extensive training from scratch. Extensive experiments confirm GSA's superior performance in scenarios requiring in-context recall and in T2R settings.

## 1   Introduction

Transformers [88] have emerged as the predominant architecture for *most, if not all*, sequence modeling tasks. Nevertheless, the quadratic complexity of $\mathrm{softmax}$-based standard attention (SA) poses significant challenges for long sequence modeling (e.g., video understanding and biological sequence modeling). In the context of language modeling, where sequence lengths are moderate, training efficiency is generally not a primary concern. However, during inference, the Key-Value (KV) cache [34, 64] grows linearly with the generation length, resulting in substantial memory burdens and throughput bottlenecks due to high I/O costs.

Linear (kernelized) attention [42] and its gated variants [96, 82, 68, 61, 16, 69] have received interest as promising alternatives to $\mathrm{softmax}$ attention. These models demonstrate strong performance in language modeling and understanding tasks. Notably, they can be reframed as RNNs during inference, achieving constant memory complexity and thereby significantly enhancing inference efficiency.

However, two key issues persist with these models: (i) Performance-wise, recent research indicates that linear recurrent models still struggle with tasks requiring in-context retrieval or learning [2, 1, 37,

---

*Equal contributions. Work was conducted during Yu Zhang's internship at Tencent AI Lab.
†Corresponding authors.

38th Conference on Neural Information Processing Systems (NeurIPS 2024).

28], and there is a *fundamental* recall-memory trade-off [3, 91] where all inference-time-constant-memory models face inherent limitations. (ii) In terms of training efficiency, while linear attention supports hardware-efficient chunkwise training [96] as implemented in FlashLinearAttention (FLA [95]), training from scratch on trillions of tokens remains prohibitively expensive. A paradigm, "*finetuning pretrained Transformers to RNNs*" (short for T2R [41]), has recently gained great attention [101, 10, 54, 13, 7, 90]. This approach circumvents the high cost of training from scratch by requiring only a few billion tokens for finetuning—about 1–3% of the total cost. However, linear attention uses a different kernel method from $\mathrm{softmax}$, leading to performance discrepancies when finetuning pretrained $\mathrm{softmax}$ attention models to linear attention [101].

To address these issues, we revisit the Attention with Bounded-Memory Control (ABC) model [63], which retains the $\mathrm{softmax}$ operation, thereby reducing training-finetuning discrepancies between standard and linear attention, making it ideal for T2R settings. Additionally, ABC enables more effective state utilization, requiring less state size to achieve similar performance, as observed in Peng et al. [63]. This results in more efficient inference and potentially expands the Pareto frontier of the recall-memory tradeoff [3]. However, ABC has not gained significant attention due to its mediocre language modeling performance and slow training speed.

In this work, we first reformulate ABC as two-pass linear attention linked via $\mathrm{softmax}$, allowing us to leverage the hardware-efficient chunkwise implementation from FLA [95] for more efficient training. We then identify several limitations of ABC and propose a new model, dubbed Gated Slot Attention (GSA), which is essentially a gated version of ABC, following the recent trend of enhancing linear attention with gating mechanisms [96, 69, 61, 16, 6, 62, 52, 65].

Our extensive evaluation shows that GSA not only matches performance in language modeling and understanding tasks but also significantly outperforms other linear models in *in-context recall-intensive* tasks [3, 4], without requiring a large state size like RetNet [82] or GLA [96]. In the T2R finetuning setting, we found that finetuning Mistral-7B [39] to GSA surpasses large recurrent language models (e.g., RWKV6-7B, Mamba-7B) and also outperforms finetuning Mistral-7B to other linear models (e.g., RetNet, GLA) and other T2R methods like SUPRA [54], verifying the importance of retaining the $\mathrm{softmax}$ operator. Finally, we remark that GSA achieves similar training speeds to GLA while offering an inference speedup due to its smaller state size.

## 2 Background and Preliminary

### 2.1 Transformers as Unbounded Key-Value Memories

Given $\mathbf{X} = [\boldsymbol{x}_1, \ldots, \boldsymbol{x}_T]^\top \in \mathbb{R}^{T \times d}$, where $T$ is the sequence length and $\boldsymbol{x}_i \in \mathbb{R}^d$ is the $i$-th input vector with $d$ dimensions, SA with causal masking computes the output matrix:

$$\mathbf{O} = f((\mathbf{Q}\mathbf{K}^\top) \odot \mathbf{M})\mathbf{V}, \tag{1}$$

where $\mathbf{Q}, \mathbf{K}, \mathbf{V} \in \mathbb{R}^{T \times d}$ are linear mappings of the input $\mathbf{X}$ via learnable weights $\mathbf{W}_q, \mathbf{W}_k, \mathbf{W}_v \in \mathbb{R}^{d \times d}$, $\mathbf{M} = \{M_{ij} = 1 \text{ if } i \geq j \text{ o.w. } -\infty\}$ is the causal mask to prevent future information leakage, $\odot$ denotes element-wise production, and $f(\cdot)$ is $\mathrm{softmax}(\cdot)$.

Generally, $\mathbf{K}, \mathbf{V}$ can be viewed as neural *key-value memories* $\widetilde{\mathbf{K}}_t, \widetilde{\mathbf{V}}_t \in \mathbb{R}^{m \times d}$, respectively [81, 24], where $m$ is the number of memory slots. At step $t$, the query $\boldsymbol{q}_t = \mathbf{W}_q \boldsymbol{x}_t \in \mathbb{R}^d$ first attends to the key memories $\widetilde{\mathbf{K}}_t$ to retrieve relevant information, which is then summarized into $\boldsymbol{o}_t$ by computing a weighted sum of the value memories $\widetilde{\mathbf{V}}_t$ [104], where the weights are the normalized attention scores:

$$\boldsymbol{o}_t = \widetilde{\mathbf{V}}_t^\top f(\widetilde{\mathbf{K}}_t \boldsymbol{q}_t). \tag{2}$$

From this perspective, Transformers are equipped with an unbounded number of memory slots, which grow linearly with respect to the sequence length [57] (i.e., $m = t$ for step $t$)—a new key $\boldsymbol{k}_t = \mathbf{W}_k \boldsymbol{x}_t \in \mathbb{R}^d$ is assigned with a unique memory slot upon its introduction. This leads to a simple memory updating rule: $\widetilde{\mathbf{K}}_t = \widetilde{\mathbf{K}}_{t-1} \cup \{\boldsymbol{k}_t\}$. The value memories $\widehat{\mathbf{V}}_t$ are updated in a similar way. This mechanism, however, comes at the cost of quadratic time complexity in terms of the sequence length for training and $O(Td)$ time/memory complexity for inference [64], posing challenges for large-scale models.

## 2.2 ABC [63]: Linearizing Attention with Bounded Memory Control

From a key-value memory perspective, the training and inference complexity of self-attention (SA) can be reduced by fixing the number of memory slots to a constant size $m \ll T$ [27, 51, 63]. One straightforward way to achieve this is by employing a *first-in-first-out* memory management strategy, commonly known as sliding window attention (SWA). However, SWA is inefficient because it discards all information outside the window, leading to poor performance in balancing the recall-memory tradeoff [3]. To achieve acceptable performance, SWA often requires a large window size (e.g., 4,096 tokens in Mistral [39]), which diminishes its advantage over to global attention.

When the number of tokens in a sequence exceeds the number of memory slots, it becomes necessary to store information from multiple tokens in a single slot. To address this challenge, Peng et al. [63] propose the Attention-with-Bounded-memory-Control (ABC) mechanism, which allows multiple tokens to be written into a single slot:

$$\widetilde{\mathbf{K}}_t = \widetilde{\mathbf{K}}_{t-1} + \boldsymbol{\phi}_t \otimes \boldsymbol{k}_t \in \mathbb{R}^{m \times d}, \quad \widetilde{\mathbf{V}}_t = \widetilde{\mathbf{V}}_{t-1} + \boldsymbol{\phi}_t \otimes \boldsymbol{v}_t \in \mathbb{R}^{m \times d}, \quad \mathbf{o}_t = \widetilde{\mathbf{V}}^T f(\widetilde{\mathbf{K}}_t^T \mathbf{q}_t) \in \mathbb{R}^d \quad (3)$$

where

$$\boldsymbol{\alpha}_i = \exp\left(\mathbf{W}_\phi \mathbf{x}_i\right) \in \mathbb{R}^m, \quad \boldsymbol{\phi}_i = \frac{\boldsymbol{\alpha}_i}{\sum_{j=1}^i \boldsymbol{\alpha}_j} \in (0,1)^m \quad (4)$$

Here, $(\boldsymbol{\phi}_i)_j$ represents the writing intensity of the $i$th token to the $j$th slot, obtained using a cumulative softmax function (cf. [63, footnote 5]), which can be computed with a prefix sum.

**ABC as two-pass linear attention.** The outer-product-based additive memory update rule in Eq. 3 bears a resemblance to linear attention [42], which involves the following recurrence[3]:

$$\mathbf{S}_t = \mathbf{S}_{t-1} + \boldsymbol{k}_t \otimes \boldsymbol{v}_t \in \mathbb{R}^{d \times d}, \qquad \boldsymbol{o}_t = \mathbf{S}_t^T \boldsymbol{q}_t \in \mathbb{R}^d \quad (5)$$

We denote this linear attention operator that computes $\boldsymbol{o}_i$ from $\boldsymbol{q}_i, \boldsymbol{k}_i$ and $\boldsymbol{v}_i$ (Eq. 5) by $\{\boldsymbol{o}_i\}_{i=1}^T = \mathrm{LA}(\{\boldsymbol{q}_i, \boldsymbol{k}_i, \boldsymbol{v}_i\}_{i=1}^T)$. We show that the ABC operations can be written as

$$\{\boldsymbol{o}_i'\}_{i=1}^T = \mathrm{LA}(\{\boldsymbol{q}_i, \boldsymbol{k}_i, \boldsymbol{\phi}_i\}_{i=1}^T),$$
$$\{\boldsymbol{o}_i\}_{i=1}^T = \mathrm{LA}(\{\mathrm{softmax}(\boldsymbol{o}_i'), \boldsymbol{\phi}_i, \boldsymbol{v}_i\}_{i=1}^T),$$

where $\boldsymbol{o}_i' \in \mathbb{R}^m, \boldsymbol{o}_i \in \mathbb{R}^d$. Therefore, ABC can enjoy hardware-efficient linear-time chunkwise training [96], as implemented in the FLA library [95].

**Remarks on state size.** Peng et al. [63] empirically demonstrated that ABC requires a smaller state size to achieve comparable performance to other linear attention models, resulting in improved inference efficiency. We offer the following intuitive explanation: the new query $\boldsymbol{o}'$ aggregates the entire history through the initial pass of linear attention, making it more context-aware and better at locating desired items for retrieval. The subsequent softmax operator helps mitigate the attention dilution issue [66]. From the perspective of Hopfield networks, softmax can exponentially increase the memory size [45]. Together, these factors suggest that ABC may possess an implicit large memory capacity, even with a small actual recurrent state size.

## 2.3 GLA [96]: Linear Attention with Gating Mechanism

Linear attentions underperform softmax-attention Transformers in language modeling by a notable margin. RetNet [82] and TransnormerLLM [68] incorporate a *data-independent* exponential decay factor for memory update as

$$\mathbf{S}_t = \gamma \mathbf{S}_{t-1} + \boldsymbol{k}_t \otimes \boldsymbol{v}_t \in \mathbb{R}^{d \times d},$$

where $\gamma \in (0,1)$ is a scalar data-independent decaying factor; that is, the decay rate is fixed across time steps and hidden channels (under the same head), disrespect to the input tokens. RetNet has shown better language modeling performance compared to vanilla linear attentions thanks to the decaying mechanism.

---

[3]For simplicity, we omit the normalization term, which has been shown to be unnecessary [75, 66, 52, 82, 96].

However, research in recurrent neural networks (RNNs) has shown that *data-dependent* decay (or forget gates) is crucial for selectively retaining and forgetting information [22, 26], thus better leveraging the fixed recurrent hidden state. This selective mechanism has been revisited in recent state-space models [29, 16]. Inspired by LSTMs, Gated Linear Attention (GLA) [52, 96] introduces data-dependent decay parameters $\mathbf{G}_t \in (0,1)^{d \times d}$ to gate the hidden state as follows,

$$\mathbf{S}_t = \mathbf{G}_t \odot \mathbf{S}_{t-1} + \boldsymbol{k}_t \otimes \boldsymbol{v}_t \in \mathbb{R}^{d \times d}, \quad \boldsymbol{o}_t = \mathbf{S}_t^T \boldsymbol{q}_t \in \mathbb{R}^d.$$

[96] show that if gates are parameterized in an outer product form $\mathbf{G}_t = \boldsymbol{\alpha}_t \otimes \boldsymbol{\beta}_i$, and $\boldsymbol{\alpha}_t, \boldsymbol{\beta}_t \in [0,1]^d$ depend solely on input $\boldsymbol{x}_t$, such recurrence can be rewritten as matrix multiplication, allowing for hardware-efficient training with a chunkwise parallel form. In what follows, we will use the following notation $\text{GLA}(\{\boldsymbol{q}_i, \boldsymbol{k}_i, \boldsymbol{v}_i, \boldsymbol{\alpha}_i, \boldsymbol{\beta}_i\}_{i=1}^T) = \{\boldsymbol{o}_i\}_{i=1}^T$ to denote this computation. It is common to set $\boldsymbol{\beta}_i = \mathbf{1}$ as in [96, 69, 61], which is also often written in the following equivalent form:

$$\mathbf{S}_t = \text{Diag}(\boldsymbol{\alpha}_t)\mathbf{S}_{t-1} + \boldsymbol{k}_t \otimes \boldsymbol{v}_t.$$

Here $\boldsymbol{k}_t$ can be viewed as the input gate, and $\boldsymbol{\alpha}_t$ can be viewed as the forget gate. In gated RNN literature, it is common to couple these two gates via $\boldsymbol{k}_t = 1 - \boldsymbol{\alpha}_t$ [12, 106, 67]. In particular, Qin et al. [69] proposed HGRN2, which uses this strategy as an improved parameterization of GLA, showing better performance in language modeling.

# 3  Method

## 3.1  Motivation: Issues with ABC

We identify two primary limitations in ABC's memory update rule. Firstly, it lacks a forgetting mechanism, resulting in indefinite retention of items once written into memory slots. This prevents efficient memory reuse by impeding the prompt clearance of slots for new information.

Secondly, the rule introduces an unwarranted inductive bias favoring tokens at the sentence's beginning. This contradicts the recency bias in natural language, where more recent information is often more relevant. Prioritizing initial tokens over the recent ones conflicts with this inherent tendency in natural language processing.

Specifically, for the first token, the writing strength to all slots is maximized (i.e., $\phi_1 = \mathbf{1} \in \mathbb{R}^m$), causing every memory slot to retain a copy of the first token's representation. The absence of a forgetting mechanism exacerbates this issue. For subsequent tokens, the writing strength diminishes due to the influence of earlier tokens, as a result of the cumulative $\text{softmax}$ in Eq. 4. This makes it challenging for the model to retain later tokens without learning a significantly large $\alpha_i$, potentially leading to instability in long-context settings, as observed by Zhang et al. [100].

## 3.2  Gated Slot Attention (GSA): ABC with gating mechanism

To address these limitations, we propose Gated Slot Attention (GSA), which incorporates a gating mechanism to simultaneously resolve both issues by: (i) enabling the forgetting of historical information, and (ii) introducing a recency inductive bias, as detailed below.

For each memory slot, the update rule is a simple gated RNN with a scalar data-dependent gating value $\alpha_i \in [0,1]$,

$$(\widetilde{\mathbf{K}}_t)_i = \alpha_i(\widetilde{\mathbf{K}}_{t-1})_i + (1-\alpha_i)\boldsymbol{k}_t \in \mathbb{R}^d, \qquad (\widetilde{\mathbf{V}}_t)_i = \alpha_i(\widetilde{\mathbf{V}}_{t-1})_i + (1-\alpha_i)\boldsymbol{v}_t \in \mathbb{R}^d$$

and these can be written in matrix form, which is reminiscent of HGRN2 [69].

$$\widetilde{\mathbf{K}}_t = \text{Diag}(\boldsymbol{\alpha}_t) \cdot \widetilde{\mathbf{K}}_{t-1} + (1-\boldsymbol{\alpha}_t) \otimes \boldsymbol{k}_t \in \mathbb{R}^{m \times d}$$
$$\widetilde{\mathbf{V}}_t = \text{Diag}(\boldsymbol{\alpha}_t) \cdot \widetilde{\mathbf{V}}_{t-1} + (1-\boldsymbol{\alpha}_t) \otimes \boldsymbol{v}_t \in \mathbb{R}^{m \times d} \tag{6}$$
$$\boldsymbol{o}_t = \widetilde{\mathbf{V}}^T \text{softmax}(\widetilde{\mathbf{K}}_t^T \mathbf{q}_t) \in \mathbb{R}^d$$

**GSA as two-pass GLA.** It is straightforward to see that we can write GSA as a two-pass GLA as shown below:

$$\{\boldsymbol{o}_t'\}_{t=1}^T = \text{GLA}\left(\{\boldsymbol{q}_t, \boldsymbol{k}_t, 1-\boldsymbol{\alpha}_t, \boldsymbol{\alpha}_t, \mathbf{1}\}_{t=1}^T\right)$$
$$\{\boldsymbol{o}_t\}_{t=1}^T = \text{GLA}\left(\{\text{softmax}(\boldsymbol{o}_t'), 1-\boldsymbol{\alpha}_t, \boldsymbol{v}_t, \mathbf{1}, \boldsymbol{\alpha}_t\}_{t=1}^T\right) \tag{7}$$

Therefore, we can adapt GLA's hardware-efficient chunkwise training algorithm for GSA training, as shown in § A and § B. We illustrate the recurrent representation of GSA in Figure 1.

## 3.3 Neural Architecture

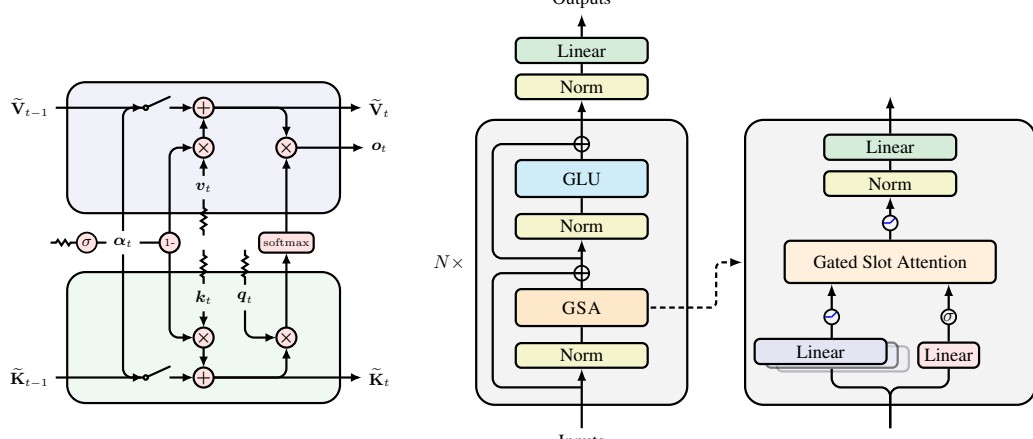

Figure 1: The recurrent representation of GSA. 〰️ means taking $\boldsymbol{x}_t$ as input.

Figure 2: The backbone of our proposed GSA models.

The overall architecture of our proposed model, GSA, is shown in Figure 2. Following the Llama architecture [86], we use a stack of $L$ GSA blocks, each comprising a GSA token mixing layer followed by a Gated Linear Unit (GLU) channel mixing layer [19, 33].

We utilize the multi-head attention mechanism [88] to capture different aspects of the input. For each head $h$, the input to GSA token mixing is defined as

$$\boldsymbol{q}_i^h, \boldsymbol{k}_i^h, \boldsymbol{v}_i^h = \phi(\mathbf{W}_q^h \boldsymbol{x}_i), \phi(\mathbf{W}_k^h \boldsymbol{x}_i), \phi(\mathbf{W}_v^h \boldsymbol{x}_i) \tag{8}$$

where $\phi$ is the Swish activation following [68]. The forget gate is obtained by a linear transformation followed by a sigmoid activation $\sigma$ with a damping factor $\tau$ [96, 83]: $\boldsymbol{\alpha}_i^h = \sigma(\mathbf{W}_\alpha^h \boldsymbol{x}_i)^{1/\tau}$, [4] where the damping factor is to regulate the forget gate value to one, which has been shown to be crucial for long-term dependency modeling [30, 67]. We feed them into a GSA layer to obtain outputs as described in Eq. 7:

$$\{\boldsymbol{o}_i^h\}_{i=1}^T = \text{GSA}(\{\boldsymbol{q}_i^h, \boldsymbol{k}_i^h, \boldsymbol{v}_i^h, \boldsymbol{\alpha}_i^h\}_{i=1}^T)$$

Finally, we obtain output via

$$\boldsymbol{y_i} = \mathbf{W}_o \left( \text{RMSNorm} \left( \text{Swish} \left( \text{Concat} \left( \boldsymbol{o}_i^1, \cdots, \boldsymbol{o}_i^H \right) \right) \right) \right) \tag{9}$$

The total number of parameters for $\mathbf{W}_q, \mathbf{W}_k, \mathbf{W}_v$, and $\mathbf{W}o$ is already $4d^2$, which is the same as in a single standard softmax-attention layer. To control the overall parameter count, we aim to keep the parameters for $\mathbf{W}_\alpha$, which amount to $dHm$, relatively small. In practice, we set $m = 64$ to achieve a balance between efficiency and effectiveness (§ 4.1.4). One way to further manage the total parameter count is by reducing the number of heads. In practice, we set $H = 4$, ensuring that $Hm \ll d$. This keeps the total number of parameters approximately equal to $4d^2$. [5]

## 4 Experiments

### 4.1 Language Modeling

We perform moderate-scale language modeling experiments with 1.3B and 2.7B parameters on Slimpajama corpus [79] for 100B tokens each.

We compare the performance of GSA against Llama Transformer architecture (i.e., Xfmr++ [86] and recent subquadratic architectures including: Mamba [29], RetNet [82], GLA [96] and HGRN2 [69]. We refer readers to § C for more details on baselines and other experimental setups.

---

[4] In practice we set $\tau = 8$.

[5] For instance, in a 1.3B model with $H \times m = 64 \times 4 = 256$ and $d = 2,048$, the total number of parameters amount to $4.125d^2$, introducing only a $0.125d^2$ overhead.

### 4.1.1 Results on commonsense reasoning tasks

Following [29, 96], we report the perplexities and zero-shot performance of commonsense reasoning tasks including $ARC_e$ & $ARC_c$ (ARC-easy, ARC-challenge) [14]; Hella. (Hellaswag) [99], Lamb. (Lambada) [59], PIQA [8], Wiki. (Wikitext) [55], and Wino. (Winograde) [73]. We note that these tasks are typically short in length and do not require in-context learning capabilities, thus they do not adequately reflect long-context modeling or in-context learning retrieval abilities. Nevertheless, as shown in Table 1, we found that GSA performs comparably to the recent strong model HGRN2 with an equally sized hidden state, while outperforming GLA and RetNet even with a smaller state size.

Table 1: The zero-shot results of 1.3B and 2.7B models evaluated by `lm-evaluation-harness` [21]. $L$ denotes number of layer while $d$ denotes the model dimension.

| | State size | Lamb. ppl↓ | Wiki. ppl↓ | $ARC_e$ acc | $ARC_c$ acc_n | Hella. acc_n | Lamb. acc | PIQA acc | Wino. acc | Avg. |
|---|---|---|---|---|---|---|---|---|---|---|
| *1.3B parameters with 100B training tokens, L=24, d=2,048* | | | | | | | | | | |
| Xfmr++ | N/A | 15.3 | 17.1 | 54.1 | 27.1 | 49.3 | 47.0 | 70.3 | **54.9** | 50.5 |
| Mamba | $64 \times Ld$ | 15.4 | 17.3 | 57.1 | 28.2 | 50.3 | 44.4 | 71.8 | 52.3 | 50.7 |
| RetNet | $512 \times Ld$ | 15.4 | 17.3 | 57.4 | 27.9 | 50.3 | 44.6 | 71.7 | 51.8 | 50.6 |
| GLA | $256 \times Ld$ | 15.4 | 17.6 | 55.4 | 27.7 | 49.0 | 46.4 | 69.9 | 54.0 | 50.4 |
| HGRN2 | $128 \times Ld$ | **11.8** | 16.9 | **58.1** | 28.1 | **51.8** | **49.4** | 71.4 | 52.3 | **51.9** |
| GSA | $128 \times Ld$ | 12.6 | **16.7** | **58.1** | **28.2** | 51.0 | 47.4 | **72.0** | 53.4 | 51.7 |
| *2.7B parameters with 100B training tokens, L=32, d=2,560* | | | | | | | | | | |
| Xfmr++ | N/A | 10.7 | 15.2 | 59.8 | 27.5 | 54.2 | 52.3 | 72.7 | **56.2** | 53.8 |
| Mamba | $64 \times Ld$ | 13.6 | 15.9 | 60.7 | 29.8 | 53.9 | 46.4 | 72.8 | 53.9 | 52.9 |
| RetNet | $512 \times Ld$ | 11.9 | 15.8 | 59.6 | 28.1 | 54.0 | 49.6 | 72.3 | 53.8 | 52.9 |
| GLA | $256 \times Ld$ | 12.4 | 15.5 | 59.2 | 29.9 | 54.0 | 50.4 | 71.7 | 55.7 | 53.5 |
| HGRN2 | $128 \times Ld$ | **8.8** | **14.6** | 60.8 | 30.3 | **58.7** | **55.4** | 73.0 | 54.2 | **55.4** |
| GSA | $128 \times Ld$ | 9.8 | 14.8 | **61.9** | **30.7** | 57.0 | 52.7 | **73.5** | 56.0 | 55.3 |

### 4.1.2 Results on in-context recall-intensive tasks

While subquadratic models can achieve comparable performance to (softmax-based) Transformers in language modeling and understanding tasks, their performance on recall-intensive tasks significantly lags behind Transformers and varies greatly across different subquadratic models, as observed in many recent studies [3, 4, 96, 97]. Therefore, it is crucial to improve linear models on in-context recall-intensive tasks.

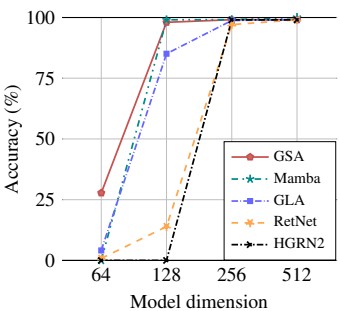

(a) Results on the synthetic MQAR task. We adopt the most challenging settings in [2], utilizing a sequence length of 512 and 64 key-value pairs. Xfmr++ with standard attention achieves near-perfect results in this settings and is thus omitted for brevity.

(b) Results on the recall-intensive tasks used in [4]. We truncate the input to a maximum of 2K tokens.

| | State size | FDA | SWDE | SQuAD | NQ | TriviaQA | Drop | Avg. |
|---|---|---|---|---|---|---|---|---|
| *1.3B params / 100B tokens, L=24, d=2048* | | | | | | | | |
| Xfmr++ | N/A | 46.0 | 29.2 | 41.0 | 24.8 | 58.8 | 21.3 | 36.9 |
| Mamba | $64 \times Ld$ | 13.9 | 25.4 | 33.2 | 18.5 | 53.5 | **21.7** | 27.7 |
| RetNet | $512 \times Ld$ | 21.2 | 27.2 | 34.0 | 15.5 | 52.7 | 20.0 | 28.4 |
| GLA | $256 \times Ld$ | **26.7** | **30.6** | 34.8 | 21.5 | 56.0 | 19.1 | 31.4 |
| HGRN2 | $128 \times Ld$ | 9.9 | 23.1 | 32.0 | 16.4 | 55.2 | 19.1 | 25.9 |
| GSA | $128 \times Ld$ | 23.6 | 29.8 | **36.0** | **23.2** | **57.0** | 20.9 | **31.8** |
| *2.7B params / 100B tokens, L=32, d=2560* | | | | | | | | |
| Xfmr++ | N/A | 62.3 | 30.9 | 44.3 | 29.3 | 61.8 | 21.4 | 41.7 |
| Mamba | $64 \times Ld$ | 21.5 | 26.7 | 34.2 | 21.2 | 57.0 | **22.2** | 30.5 |
| RetNet | $512 \times Ld$ | 24.1 | 26.1 | 36.4 | 20.4 | 57.3 | 21.8 | 31.0 |
| GLA | $256 \times Ld$ | 30.3 | **35.5** | 36.8 | 23.3 | 58.2 | 21.8 | 34.3 |
| HGRN2 | $128 \times Ld$ | 15.0 | 29.9 | 35.1 | 17.0 | 59.8 | 20.0 | 29.5 |
| GSA | $128 \times Ld$ | **39.1** | 33.5 | **39.0** | 26.9 | 60.8 | 19.9 | **36.5** |

**MQAR.** We first present the results on the multi-query associative recall (MQAR) task [2], a diagnostic synthetic task that requires models to retrieve multiple associative key-value pairs from the context. This task has been shown to strongly correlate with language modeling performance [2]. The results in Table 3a validate the effectiveness of GSA.

**Real-world tasks.** Next, we evaluate the zero-shot in-context learning performance on recall-intensive tasks, as used in Arora et al. [4].[6] Specifically, we assess information retrieval on FDA [93] and SWDE [49], which are designed to evaluate retrieval from in-context passages scraped from HTML/PDFs. We also evaluate question answering on SQuAD [70], NQ [46], TriviaQA [40], and Drop [20], where models must ground their answers in in-context documents.

As shown in Table 3b, Xfmr++ achieves the best average performance, as expected. Meanwhile, GSA outperforms all other subquadratic baseline models by a notable margin without requiring a larger state size. We believe this advantage stems from GSA's context-aware memory readout mechanism (as discussed in §2.2) and its forgetting mechanism (i.e., the gating mechanism), enabling it to manipulate finite-sized memory more effectively.

Table 2: Ablation study results for 340M models trained on 10B Slimpajama tokens.

|  | PPL ($\downarrow$) |
| --- | --- |
| GSA w/ 64 slots | 13.51 |
| *Ablations on gating mechanism* | |
| w/o decay (i.e., ABC) | 16.94 |
| w/ data-independent decay | 15.83 |
| *Ablations on non-linearity* | |
| $-$ softmax | 14.03 |
| $-$ softmax $+$ Swish | 13.71 |
| $-$ softmax $+$ ReLU | 13.69 |
| $-$ softmax $+$ ReLU$^2$ | 13.95 |
| *Ablations on slot size* | |
| w/ 32 slots | 13.74 |
| w/ 128 slots | **13.46** |

### 4.1.3 Ablation

Table 2 presents the results of our ablation studies. Our findings indicate that: (i) the inclusion of the gating mechanism in GSA is crucial for improving language modeling perplexity; (ii) applying softmax non-linearities after the first recurrent pass is beneficial; and (iii) using 64 slots strikes an optimal balance between performance and efficiency. [7]

### 4.1.4 Efficiency

Fig. 4a illustrates the training throughput for four models on a single H800 GPU[8]. To optimize memory usage, we employ the technique of recomputing the recurrent hidden state during the backward pass, as done in FLA [95] and Mamba2 [16]. This approach results in reduced memory consumption (Fig. 4b) at the cost of slightly lower training throughputs (Fig. 4a).

Despite requiring two GLA passes, GSA maintains comparable training throughputs to GLA due to its reduced state size. Since inference is primarily memory-bound, inference speed highly correlates with state size. As a result, GSA, with its smaller state size compared to RetNet and GLA, achieves faster inference speeds, as shown in Figure 4c.

### 4.2 Finetuning Pretrained Transformers to RNNs

The concept of finetuning pretrained Transformers to linear Transformers for recurrent inference was first introduced in T2R [41]. This approach uses pretrained language model weights to initialize all parameters, leveraging the similarity between linear attention and softmax attention, and finetunes all parameters, significantly reducing the total training time compared to training from scratch. Kasai et al. [41] also introduced a *parametric* feature map, implemented as a learnable MLP layer followed by ReLU, applied after the query/key projections. SUPRA, a follow-up to T2R, found that the original T2R approach did not perform well in the era of LLMs, and highlighted the importance of

---

[6]Since our pretrained models are neither instruction-tuned nor instruction-aligned, following Arora et al. [4], we use their Cloze Completion Formatting prompts for evaluation. It is noteworthy that results for certain tasks may differ significantly from those obtained using `lm-evaluation-harness` [21] due to variations in prompt templates.

[7]Empirically, we found that 32, 64, and 128 slots result in training throughputs of 46.7K, 44.1K, and 37.1K tokens/s, respectively, under the settings described in the next section. Given the marginal improvement when increasing the slot size from 64 to 128, along with the significant slowdown in training, we chose 64 slots.

[8]We utilize the training throughput benchmark scripts provided by FLA [95] for our measurements.

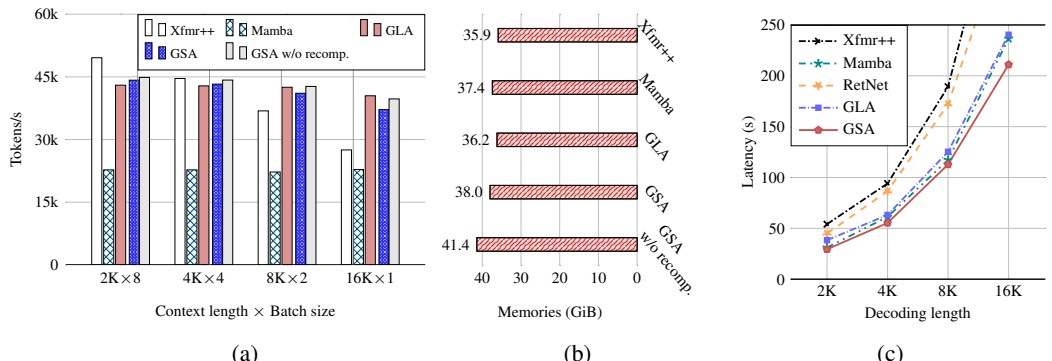

Figure 4: (a) Training throughput of various 1.3B models on a single H800 GPU, with a fixed batch size containing 16K tokens. "GSA w/o recomp." indicates the use of the GSA kernel without hidden state recomputation during the backward pass. (b) Memory footprint (in GiB) of each 1.3B model during training with a batch size containing 16K tokens. (c) Inference latency (in seconds) of each 1.3B model on a single H800 GPU with 2K prefix tokens and a batch size of 1.

Table 3: Performance comparison across various 7B models. ♣ denotes models using softmax-attention. † denotes our results.

| Shot(s) | Size | Tokens | ARC$_e$ 0 | ARC$_c$ 0 | Hella. 0 | PIQA 0 | Wino. 0 | NQ 5 | TriviaQA 5 | BBH 3 | MMLU 5 | Avg. |
|---|---|---|---|---|---|---|---|---|---|---|---|---|
| *Models trained from scratch (for reference)* | | | | | | | | | | | | |
| RWKV6 | 7B | 1.4T | 73.6 | 44.0 | 75.2 | 78.4 | 68.5 | 20.9 | 59.5 | 23.4 | 43.9 | 54.1 |
| Mamba | 7B | 1.2T | 77.6 | 46.8 | 77.8 | 81.0 | 72.3 | 25.4 | 66.2 | 21.5 | 33.2 | 55.7 |
| Llama2♣ | 7B | 2T | 76.4 | 46.2 | 76.0 | 78.0 | 69.2 | 26.0 | 64.2 | 39.1 | 45.5 | 57.8 |
| Gemma♣ | 7B | 6T | 81.5 | 53.2 | 80.5 | 79.8 | 74.0 | 24.3 | 63.7 | 58.9 | 63.2 | 64.3 |
| Mistral♣ | 7B | ? | 80.8 | 54.0 | 81.1 | 80.6 | 74.0 | 29.7 | 70.3 | 56.5 | 62.4 | 65.5 |
| *Models finetuned from Mistral 7B* | | | | | | | | | | | | |
| SUPRA | 7B | +20B | 74.6 | 42.3 | 74.8 | **80.1** | 67.4 | - | - | - | 28.0 | - |
| RetNet† | 7B | +20B | 73.3 | 39.9 | 72.9 | 77.8 | 66.1 | 16.2 | 43.0 | 8.7 | 26.1 | 47.1 |
| GLA† | 7B | +20B | 74.6 | **44.0** | 75.9 | 79.2 | 69.5 | 22.2 | 57.8 | 20.8 | 28.4 | 52.5 |
| GSA† | 7B | +20B | **75.9** | 43.9 | **76.5** | 78.7 | **70.1** | **23.4** | **60.7** | **23.5** | **32.4** | **53.9** |
| SUPRA | 7B | +100B | **76.0** | 45.7 | 77.1 | **79.9** | 70.3 | 24.7 | 60.4 | 19.8 | 34.1 | 54.2 |
| GSA† | 7B | +100B | **76.0** | **46.9** | **77.9** | 78.9 | **72.6** | **26.9** | **65.8** | **29.3** | **38.1** | **56.9** |

output normalization and a decay mechanism—adopted from RetNet [82]—as critical for finetuning performance. As a result, SUPRA essentially combines T2R and RetNet by finetuning pretrained Transformers into a RetNet architecture, though it excludes the Swish output gate.

**Settings.** In our preliminary experiments, we found that the learnable MLP layer was unnecessary and could be merged into the query and key projections, similar to the approach in Peng et al. [63]. We finetuned the pretrained Transformer Mistral 7B [39] to RetNet, as well as to GLA and GSA models. Following SUPRA, we add ReLU as the feature map activation for RetNet and GLA, which originally used an identity feature map without activation [9], and also excluded the Swish output gate. For RetNet, there were no additional parameters; for GLA, the low-rank forget gate, and for GSA, the $W_\alpha$ matrix are trainable parameters, though both are small in parameter count and negligible in terms of the total model size. We set the peak learning rate to $3 \times 10^{-5}$ with 1K steps of linear warmup following SUPRA. The training length was set to 2K tokens, with a batch size of 2M tokens. For convenience, we trained on the SlimPajama corpus, while SUPRA used RefineWeb [60], a higher-quality corpus. We leave the use of RefineWeb for future work.

---

[9]However, SUPRA reported poor performance with this strategy due to a significant discrepancy between training and finetuning, where an identity map can lead to negative attention scores, a pattern unseen in pretrained Transformers due to the nonnegativity of softmax.

**Main results.** Following Jiang et al. [39], Touvron et al. [87], we evaluated the models on common-sense reasoning tasks: $ARC_e$ and $ARC_c$ [14], Hellaswag [99], PIQA [8], and Winogrande [73]; world knowledge tasks: NQ [46] and TriviaQA [40]; and popular aggregated benchmarks: MMLU [31] and BBH [85]. Results are shown in Table 3. We observed a clear advantage in finetuning Mistral to GSA compared to GLA or RetNet, confirming our intuition that preserving softmax is beneficial in T2R settings. When trained with 100B tokens, Mistral-to-GSA outperforms RWKV6 and Mamba on average, even though those models were trained on over 1T tokens, thereby reducing the required training data size.

**Long-context ability evaluation.** Following Xiong et al. [94], we evaluated the models on long-sequence tasks, including Qasper [18], NarrativeQA [44], QuALITY [58], and QMSum [105]. For each task, the input was truncated to 16K tokens, which is $8\times$ the training length.

The results are shown in Table 4. Notably, GSA consistently outperforms other subquadratic models across all four tasks. We attribute this to the same factors observed in in-context recall-intensive task settings. Interestingly, Mistral-to-GSA also demonstrates overall better perfor-

Table 4: Long-context performance comparison.

|  | Qasper | NarrativeQA | QuALITY | QMSum |
|---|---|---|---|---|
| *Models trained from scratch (for reference)* | | | | |
| RWKV6 | 9.2 | 14.4 | 30.8 | 1.1 |
| Mamba | 5.6 | 27.9 | 27.5 | 0.8 |
| Mistral♣ | 25.8 | 25.1 | 38.0 | 5.0 |
| *Models finetuned from Mistral 7B on 20B tokens* | | | | |
| RetNet | 11.1 | 0.0 | 26.2 | 0.0 |
| GLA | 18.4 | 17.2 | 30.9 | 9.0 |
| GSA | **18.8** | **19.2** | **32.0** | **10.0** |

mance compared to RWKV6 and Mamba, which were trained from scratch on >1T token.

## 5 Related works

**Matrix-valued linear RNNs with hardware-efficient training.** Traditional RNNs (e.g., LSTM [32], GRU [12]) maintain 1-dimensional hidden states, which are often too small to capture sufficient information. Recent work emphasizes the importance of expanding the size of recurrent states [29, 69, 96, 82, 61, 16]. However, naive state expansion dramatically increases FLOPs and I/O costs, making training impractical. To address this, Mamba introduces an I/O-aware approach, reducing I/O costs by materializing parameters and hidden states only on SRAM (instead of HBM). However, Mamba's recurrence cannot be expressed in matmul form, leading to two key issues: (i) high FLOP count cannot be optimized via tensor cores (the GPU's fast matmul unit), resulting in slower runtimes; and (ii) the recurrent hidden states cannot be compactly represented and must be materialized on SRAM during backpropagation, limiting the recurrent state size due to SRAM constraints.

Mamba2 [16] addresses these limitations by adopting a linear attention [42]-like approach that enables hardware-efficient training. Linear attention expands the state using outer products, allowing for both parallel attention-style computation and recurrent inference (also known as state-space duality in Mamba2). The chunkwise algorithm interpolates between parallel and recurrent forms, enabling hardware-efficient, linear-time training [33, 82, 96]. However, vanilla linear attention underperforms softmax attention in various tasks. Recent research has explored incorporating various decay or gating mechanisms to enhance model expressiveness and performance while maintaining matmul-based parallelism and chunkwise training. These include head-wise data-independent decay [82, 68]; head-wise data-dependent decay [62, 16, 6, 83]; and channel-wise data-dependent decay [96, 52, 43, 69, 61]. GSA leverages two-pass gated linear attention to further enhance capacity while allowing hardware-efficient training.

**Fast weight RNNs.** Fast weight programming [77], a classical concept intensively investigated in deep learning [5, 103, 74, 76, 56, 75, 35, 36, 52], has been shown to be closely related to (linear) Transformers [75]. The core idea involves using a slow network to produce rapid context-dependent weight modifications for the fast network. In linear attention, the fast network is a single-layer FFN with weight matrix $\mathbf{S}_t$ (Eq. 5), while the slow networks are the query/key/value projections.

Linear attention is known to suffer from limited memory capacity [75], potentially due to the constraints of a single-layer FFN without a large representation. In contrast, ABC and GSA can be viewed as implementing a two-layer fast FFN with either additive update rule or gated update rule [74, 52], where the weight matrices are $\widetilde{\mathbf{K}}_t$ and $\widetilde{\mathbf{V}}_t$ connected by the softmax activation function (Eq. 3 and Eq. 6). This structure resembles DeltaMLP [35], which uses a delta update rule [92, 75, 98]

and a multi-layer (potentially beyond two layers) fast FFN. The greater capacity of a two-layer FFN compared to a similarly sized single-layer FFN could explain why GSA requires a smaller state size to achieve similar or even better performance, especially in long sequence and recall-intensive tasks.

**Finetuning Transformers to RNNs.** As discussed, this paradigm could significantly reduce the training cost for large-scale recurrent language models. The idea of distilling Transformers to RNNs to improve inference efficiency can be traced back to Gerstenberger et al. [23]. In the following, we briefly introduce some recent works that complement those already mentioned in §4.2 . Zhang et al. [101] highlight the desirable properties of softmax, such as attention spikiness and dot-product monotonicity, and employ a learnable MLP layer to approximate softmax behavior using logit distillation loss (while freezing other parameters). Chen et al. [10] introduce DiJiang, an effective method for approximating attention distributions using the Discrete Cosine Transform (DCT) to enable frequency-domain kernelization, leading to faster feature mapping. Bick et al. [7] propose a multi-stage distillation approach, aligning attention distributions (similar to Hedgehog [101]), hidden states, and output logits to transfer knowledge from a pretrained Transformer teacher to a student Mamba model. Wang et al. [90] distill Transformer-based LLMs into hybrid Mamba-Attention architectures in the spirit of Ren et al. [71], Lieber et al. [47], Waleffe et al. [89]. However, they freeze the FFN weights, while Choi [13] suggest that it might be more effective to unfreeze them. In this work, we highlight the importance of the softmax operator, as discussed in Zhang et al. [101], except that GSA directly incorporates softmax, while Zhang et al. [101] learns a feature map to mimic softmax, without actually including any softmax operator in the resulting model.

# 6 Limitations and future work

Due to the relatively small scale of our pretrained models (compared to large-scale models trained on trillions of tokens), we did not report any results on long-context tasks, as the performance would all be poor. However, we believe Table 4 provides positive indications of GSA's long-context capabilities, and training on a larger token horizon and with larger models would address this. For copy-oriented tasks, we observed negative results on the Phonebook Lookup [38] and Needle-In-Haystack evaluations compared to Transformers, revealing the fundamental limitations of linear recurrent models in handling "precise local token shifts and comparison", as discussed in Arora et al. [3]. Nonetheless, we expect this limitation could be significantly mitigated by pretraining a hybrid GSA-attention model, as recently explored [3, 71, 89, 47, 98], or by distilling pretrained Transformers into hybrid GSA-attention models, as in Wang et al. [90], or using different training objectives with JRT prompts, as in Arora et al. [4], or combining with YOCO [83, 25].

GSA follows GLA in using a gated update rule, although we acknowledge recent work on Parallel DeltaNet [98], which parallelizes the delta update rule computations in DeltaNet [75] over sequence length, significantly enhancing training efficiency. The delta rule is known to improve in-context retrieval ability [75, 98], aligning with one of the objectives of this work. We did not explore the analogous two-pass DeltaNet, but we leave this for future investigation, which would bring the approach closer to the original DeltaMLP [35], as discussed earlier. It would also be beneficial to compare GSA with more recent strong RNN models, such as xLSTM [6], Mamba2 [16], TTT [84], and Longhorn [48].

# 7 Conclusions

This work introduces Gated Slot Attention (GSA), which enhances ABC [63] with a gating mechanism inspired by Gated Linear Attention (GLA [96]). By framing GSA as a two-pass GLA, we can leverage hardware-efficient implementations of GLA [95] to train GSA. As such, GSA benefits from context-aware memory reading and forgetting, implicitly increasing the model's capacity despite a small actual state size, which improves training and inference efficiency. Through extensive experiments, we demonstrate the advantages of GSA in in-context recall-intensive tasks [4] and in "finetuning pretrained Transformers to RNNs" [41] scenarios.

## Acknowledgments

We would like to thank Zhen Qin and Yikang Shen for their insightful discussions, Houquan Zhou and Kazuki Irie for providing valuable feedback on this manuscript.

We gratefully acknowledge the support by LuxiTech for computational resources; and the support by the National Natural Science Foundation of China (No. 62076173, 62476187), the High-level Entrepreneurship and Innovation Plan of Jiangsu Province (No. JSSCRC2021524), and the Project Funded by the Priority Academic Program Development of Jiangsu Higher Education Institutions. Yu Zhang was partially supported by Tencent AI Lab under Wei Bi's mentorship. Songlin Yang was supported by Xianhong Wu Fellowship from MIT.

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

# A   Linear Attention and its Chunkwise Form

Linear Attention (LA) [42, 66, 68] emerges as an alternative to resolve the quadratic complexity of self-attention (SA). The key idea is to use the *kernel trick*, which replaces $\mathrm{softmax}$ with a decomposable kernel function, resulting the following *parallel form*:[10]

$$\mathbf{O} = ((\phi(\mathbf{Q})\phi(\mathbf{K})^\top) \odot \mathbf{M})\mathbf{V}. \tag{10}$$

where $\phi : \mathbb{R}^d \to \mathbb{R}^m$ functions as feature mapping applied to each input. Unfolding Eq. 10, we have

$$\boldsymbol{q}_t, \boldsymbol{k}_t, \boldsymbol{v}_t = \mathbf{W}_q\boldsymbol{x}_t, \mathbf{W}_k\boldsymbol{x}_t, \mathbf{W}_v\boldsymbol{x}_t \in \mathbb{R}^d,$$

$$\boldsymbol{o}_t = \sum_{i=1}^{t} \boldsymbol{v}_i f(\boldsymbol{k}_i^\top \boldsymbol{q}_t) = \sum_{i=1}^{t} \boldsymbol{v}_i \phi(\boldsymbol{k}_i)^\top \phi(\boldsymbol{q}_t) = \left[ \mathbf{S}_t \equiv \sum_{i=1}^{t} \phi(\boldsymbol{k}_i) \otimes \boldsymbol{v}_i \right]^\top \phi(\boldsymbol{q}_t). \tag{11}$$

$\otimes$ means outer product operation. It is clear that by leveraging the associativity, LA admits simple recurrent updating rules with matrix-valued hidden states $\mathbf{S}_t \in \mathbb{R}^{m \times d}$:

$$\boldsymbol{o}_t = \mathbf{S}_t^\top \phi(\boldsymbol{q}_t); \ \mathbf{S}_t = \mathbf{S}_{t-1} + \phi(\boldsymbol{k}_t) \otimes \boldsymbol{v}_t. \tag{12}$$

By reserving bounded $m$ memory slots only, the overall computation complexity is reduced from $O(T^2d)$ to $O(Tmd)$. When the sequence length is $T \gg m, d$, the $md$ factor has a minor impact on the complexity, and LA can be much more efficient than its counterpart with quadratic complexity.

During inference, LA enjoys the merits of RNNs, which only need to maintain $O(md)$ hidden memories, helping avoid the memory-cost KV cache management in SA mechanisms. However, Eq. 12 employs a simple additive updating rule and can be hard to "*forget*" unrelated information if necessary [62], making the limited memory states vulnerable to be chaotic.

**Gating mechanism** has played a key role in classical RNNs [32, 22, 12], which serves as a mechanism to control the information flows in the network and help read and write from the memory selectively. [82] propose to apply a *data-independent* gate to LA, significantly narrowing the gap between LA and SA: $\mathbf{S}_t = \lambda\mathbf{S}_{t-1} + \phi(\boldsymbol{k}_t) \otimes \boldsymbol{v}_t$, $\lambda \in [0, 1]$ is a non-learnable scalar. Recent work [96, 43] further imposes a finer-grained *data-dependent* gate:

$$\mathbf{S}_t = \mathrm{Diag}(\boldsymbol{\alpha}_t)\mathbf{S}_{t-1} + \phi(\boldsymbol{k}_t) \otimes \boldsymbol{v}_t, \tag{13}$$

where each $\boldsymbol{\alpha}_t \in [0, 1]^m$ from $\mathbf{A} := \{\boldsymbol{\alpha}_i\}_{i=1}^T \in [0, 1]^{T \times m}$ is dependent on the input. Alternatively, we can couple the key values with the forget gates by allowing $\phi(\boldsymbol{k}_t) = 1 - \boldsymbol{\alpha}_t$ in spirit of [12, 106] and [69], which reduces the number of parameters and improves efficiency accordingly.

## A.1   Hardware-Efficient Training

Despite the theoretical advantages of linear complexity, the recurrent form of Eq. 12 is still inefficient during training. Such recurrent computation prevents the full utilization of modern GPU parallelism over sequence lengths [53, 72]. On the other hand, the parallel form (Eq. 10) can be parallelized in similar vein as in flash attention [17, 15]. However, due to the existence of the casual mask $\mathbf{M}$, we can not rearrange its computation order by $\mathbf{KV}$ first, so that the parallel form still adheres to the quadratic complexity, which can hardly be scaled to very-long training context (e.g., sequences with more than 8K tokens).

**Chunkwise form** recurrences have been carried forward by [82], and achieve a good trade-off between the recurrent and parallel forms. [96] further disclose that the element-wise gating of Eq. 13 also satisfies the associative property required by parallel scan [9] and derive a parallelized chunkwise gated linear attention in a similar vein. The key idea is to partition the sequence into $N = \lceil \frac{T}{C} \rceil$ chunks of size $C$ with $\mathbf{Q}_{[t]} = \boldsymbol{q}_{tC}, \boldsymbol{q}_{tC+1}, \ldots, \boldsymbol{q}_{tC+C}$, and so forth for $\mathbf{K}_{[t]}, \mathbf{V}_{[t]} \in \mathbb{R}^{C \times d}, \mathbf{A}_{[t]} \in \mathbb{R}^{C \times m}$. Firstly, unrolling the $i$-th hidden state in the $t$-th chunk in Eq. 13, we get

$$\mathbf{S}_{[t],i} = \mathrm{Diag}\left(\mathbf{A}_{[t],i}\right) \mathbf{S}_{[t],i-1} + \phi\left(\mathbf{K}_{[t],i}\right) \otimes \mathbf{V}_{[t],i} = \cdots$$

$$= \mathrm{Diag}\left(\prod_{j=1}^{i} \mathbf{A}_{[t],j}\right) \mathbf{S}_{[t-1],C} + \sum_{k=1}^{i} \left(\phi(\mathbf{K}_{[t],k}) \odot \prod_{j=k+1}^{i} \mathbf{A}_{[t],j}\right) \otimes \mathbf{V}_{[t],k} \tag{14}$$

---

[10]There is a normalization term in vanilla LA similar to $\mathrm{softmax}$, [66] reveal that removing it could avoid potential gradient explosions.

We write the last hidden in the chunk $\mathbf{S}_{[t],C}$ as $\mathbf{S}_{[t]}$ interchangeably for simplicity. Define $\overrightarrow{\mathcal{A}}_{[t],i} = \prod_{j=1}^{i} \mathbf{A}_{[t],j} \in [0,1]^d$ as the cumulative decay from the start of chunk to $i$, and likewise $\overleftarrow{\mathcal{A}}_{[t],i} = \prod_{j=i+1}^{C} \mathbf{A}_{[t],j} \in [0,1]^d$ from $i+1$ to the end of the chunk, then

$$\mathbf{S}_{[t]} = \mathrm{Diag}(\overrightarrow{\mathcal{A}}_{[t],C})\mathbf{S}_{[t-1]} + (\mathbf{K}_{[t]} \odot \overleftarrow{\mathcal{A}}_{[t]})^\top \mathbf{V}_{[t]} \tag{15}$$

$\overrightarrow{\mathcal{A}}, \overleftarrow{\mathcal{A}}$ can be absorbed into $\mathbf{Q}, \mathbf{K}$ first : $\overline{\mathbf{Q}}_{[t]} = \phi(\mathbf{Q}_{[t]}) \odot \overrightarrow{\mathcal{A}}_{[t]}, \overline{\mathbf{K}}_{[t]} = \phi(\mathbf{K}_{[t]}) \odot (\overleftarrow{\mathcal{A}}_{[t]}/\overrightarrow{\mathcal{A}}_{[t],C})$. Combining them with Eq. 10 and Eq. 14, we derive the following vectorized updating rules

$$\mathbf{O}_{[t]} = \overline{\mathbf{Q}}_{[t]}\mathbf{S}_{[t-1]} + \left(\overline{\mathbf{Q}}_{[t]}\overline{\mathbf{K}}_{[t]}^\top \odot \mathbf{M}_{[t]}\right)\mathbf{V}_{[t]} \tag{16}$$

The first term is referred to as the *inter* chunk part and the second term is the *intra* chunk part. The process to get this *intra* part is a little more involved as the cumulative productions of $\overleftarrow{\mathcal{A}}_{[t]}/\overrightarrow{\mathcal{A}}_{[t],C}$ is greater than 1, which can lead to numerical instability. [96] deal with this issue by proposing a secondary-chunking strategy, and we refer readers to their paper for more details.

**Hardware considerations**   Modern GPU architectures, such as the NVIDIA A100, offer highly optimized matrix multiplication (matmul) operations through specialized Tensor Cores, achieving up to $16\times$ higher throughput than non-matmul operations [17]. However, this incurs IO overheads due to data transfer from slower, off-chip global high bandwidth memory (HBM) to on-chip shared memory (SRAM). The chunkwise form balances I/O and computation complexity tradeoffs. As shown in Eq.16, it improves parallelism over the sequence dimension while reducing non-matmul FLOPs greatly. Also, the chunk recurrent updating conducts the query and hidden states reduction in an online manner, requiring only $O(Ndm)$ hidden states materialized into HBMs, so that it can significantly reduce the memory/IO overheads. While LA enjoys much lower overall running FLOPs than SA, the chunkwise form displays a practical significant wall-clock speedup against SA, due to its hardware-efficient implementations [95].

## B   Details for GSA

Beyond the recurrent GSA form provided in Figure. 1, we give detailed, hardware-efficient procedures for the forward and backward passes of Gated Slot Attention (GSA) in Algorithm 1. For simplicity, we define $\mathbf{A} = \{\boldsymbol{\alpha}_i\}_{i=1}^T \in [0,1]^{T\times m}$, and $\mathbf{I} = \{\mathbf{1} - \boldsymbol{\alpha}_i\}_{i=1}^T \in [0,1]^{T\times m}$. The algorithm demonstrates that GSA can be modeled as a two-pass GLA, as illustrated in Fig. 5.

In the preprocessing step, we precompute the chunkwise cumulative sum of the forget gate, resulting in $\overrightarrow{\mathcal{A}}$. Subsequently, $\overrightarrow{\mathcal{A}}$ along with the queries, keys, and values are passed to engage in two

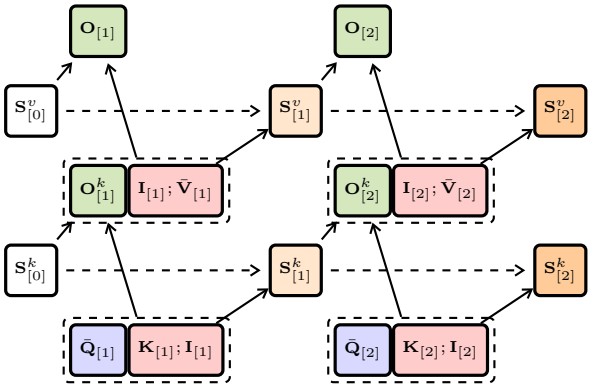

Figure 5: Diagrams of the recurrence and updating rules in Gated Slot Attention. The outputs of the first pass is taken as queries of the second pass.
☐: query nodes   ☐: key/value nodes
☐: output nodes   ☐: recurrent hidden states

GLA passes. For each chunk of size $C$, we define $\overleftarrow{\mathcal{A}}_{[i]} := \overrightarrow{\mathcal{A}}_{[i],C}/\overrightarrow{\mathcal{A}}_{[i]}$ as in Eq. 15 and Eq. 16.

In the first pass, $\overrightarrow{\mathcal{A}}, \overleftarrow{\mathcal{A}}$ is absorbed into $\mathbf{Q}, \mathbf{K}$ : $\bar{\mathbf{Q}}_{[i]} = \mathbf{Q}_{[i]} \odot \overrightarrow{\mathcal{A}}_{[i]}, \bar{\mathbf{K}}_{[i]} = \mathbf{K}_{[i]} \odot (\overleftarrow{\mathcal{A}}_{[i]}/\overrightarrow{\mathcal{A}}_{[i],C})$, then $\bar{\mathbf{Q}}$ and $\bar{\mathbf{K}}$ function as usual queries and keys, and the slot representations $\mathbf{I}$ serve as the value vectors.

$$\mathbf{O}_{[i]}^k = \underbrace{\bar{\mathbf{Q}}_{[i]}\,\mathbf{S}_{[i-1]}^k}_{\mathbf{O}_{[i]}^{\text{inter}}} + \underbrace{((\,\bar{\mathbf{Q}}_{[i]}\,\bar{\mathbf{K}}_{[i]}^\top) \odot \mathbf{M})\mathbf{I}_{[i]}}_{\mathbf{O}_{[i]}^{\text{intra}}} \in \mathbb{R}^{C\times m}$$

We use different notations from those presented in Eq.6 to enhance clarity in the chunkwise updating rules. The output $\mathbf{O}^k$ is decomposed into the *inter-chunk* recurrence and *intra-chunk* parallel computations [96].

**Algorithm 1** Hardware-Efficient Gated Slot Attention

**Define** FORWARDPASS($\mathbf{Q}, \mathbf{K}, \mathbf{V}, \mathbf{I}, \mathbf{A}$)
  Divide $\mathbf{Q}, \mathbf{K}, \mathbf{V} \in \mathbb{R}^{T \times d}, \mathbf{I}, \mathbf{A} \in \mathbb{R}^{T \times m}$
    into $N = \left\lceil \frac{T}{C} \right\rceil$ blocks           ▷ $C$ is chunk size
  **function** chunk_cumsum($\mathbf{A}$)
    **parfor** $n \leftarrow 1, N$ **do**
      Load $\mathbf{A}_{[n]}$ to SRAM
      Store $\overrightarrow{\mathcal{A}}_{[n]} \leftarrow$ cumsum($\mathbf{A}_{[n]}$) to HBM
    **return** $\overrightarrow{\mathcal{A}} \leftarrow \overrightarrow{\mathcal{A}}_{[0]}, \ldots, \overrightarrow{\mathcal{A}}_{[N]}$

  **function** gsa_fwd($\mathbf{Q}, \mathbf{K}, \mathbf{V}, \overrightarrow{\mathcal{A}}$, GATE_K)
    On chip: construct causal mask $\mathbf{M} \in \mathbb{R}^{C \times C}$
    **for** $n \leftarrow 1, N$ **do**
      Store $\mathbf{S}$ to HBM as $\mathbf{S}_{[n]}$           ▷ Initialize $\mathbf{S} = \mathbf{0}$
      Load $\mathbf{K}_{[n]}, \mathbf{V}_{[n]}, \overrightarrow{\mathcal{A}}_{[n],C} \overleftarrow{\mathcal{A}}_{[n]}$ to SRAM
      On chip: $\overleftarrow{\mathcal{A}}_{[n]} \leftarrow \overrightarrow{\mathcal{A}}_{[n],C}/\overrightarrow{\mathcal{A}}_{[n]}$
      **if** GATE_K **then**
        $\mathbf{S} \leftarrow \mathrm{Diag}(\overrightarrow{\mathcal{A}}_{[n],C})\mathbf{S} + (\mathbf{K}_{[n]} \odot \overleftarrow{\mathcal{A}}_{[n]})^\top \mathbf{V}_{[n]}$
      **else**
        $\mathbf{S} \leftarrow \mathbf{S}\mathrm{Diag}(\overrightarrow{\mathcal{A}}_{[n],C}) + \mathbf{K}_{[n]}^\top (\mathbf{V}_{[n]} \odot \overleftarrow{\mathcal{A}}_{[n]})$
    **parfor** $n \leftarrow 1, N$ **do**
      Load $\mathbf{Q}_{[n]}, \mathbf{K}_{[n]}, \mathbf{V}_{[n]}, \mathbf{S}_{[n]}, \overleftarrow{\mathcal{A}}_{[n]}, \overrightarrow{\mathcal{A}}_{[n]}$ to SRAM
      On chip:
      **if** GATE_K **then**
        $\bar{\mathbf{Q}}_{[n]} \leftarrow \mathbf{Q}_{[n]} \odot \overrightarrow{\mathcal{A}}_{[n]}$
        $\bar{\mathbf{K}}_{[n]} \leftarrow \mathbf{K}_{[n]} \odot (\overrightarrow{\mathcal{A}}_{[n]}/\overrightarrow{\mathcal{A}}_{[n],C})$
        $\mathbf{O}_{[n]} \leftarrow \bar{\mathbf{Q}}_{[n]}\mathbf{S}_{[n-1]} + \left(\mathbf{P} \equiv \bar{\mathbf{Q}}_{[n]}\bar{\mathbf{K}}_{[n]}^\top \odot \mathbf{M}\right)\mathbf{V}_{[n]}$
      **else**
        $\bar{\mathbf{V}}_{[n]} \leftarrow \mathbf{V}_{[n]} \odot (\overrightarrow{\mathcal{A}}_{[n]}/\overrightarrow{\mathcal{A}}_{[n],C})$
        $\mathbf{O}_{[n]} \leftarrow \mathbf{Q}_{[n]}\mathbf{S}_{[n-1]} + \left(\mathbf{P} \equiv \mathbf{Q}_{[n]}\mathbf{K}_{[n]}^\top \odot \mathbf{M}\right)\bar{\mathbf{V}}_{[n]}$
        $\mathbf{O}_{[n]} \leftarrow \mathbf{O}_{[n]} \odot \overrightarrow{\mathcal{A}}_{[n]}$
      Store $\mathbf{O}$ to HBM as $\mathbf{O}_{[n]}$.
    **return** $\mathbf{O}_{[1,\ldots,N]}, \mathbf{S}_{[1,\ldots,N]}$

  $\overrightarrow{\mathcal{A}} \leftarrow$ chunk_cumsum($\mathbf{A}$)           ▷ preprocessing
  $\mathbf{O}^k, \mathbf{S}^k \leftarrow$ gsa_fwd($\mathbf{Q}, \mathbf{K}, \mathbf{I}, \overrightarrow{\mathcal{A}}$, False)
  $\mathbf{Q}^v \leftarrow$ softmax($\mathbf{O}^k$)
  $\mathbf{O}, \mathbf{S}^v \leftarrow$ gsa_fwd($\mathbf{Q}^v, \mathbf{I}, \mathbf{V}, \overrightarrow{\mathcal{A}}$, True)
  **return** $\mathbf{O}$

**Define** BACKWARDPASS($\mathbf{Q}, \mathbf{K}, \mathbf{V}, \mathbf{I}, \mathbf{A}, \mathbf{O}^k, \mathrm{d}\mathbf{O}$)
  Divide $\mathbf{Q}, \mathbf{K}, \mathbf{V}, \mathbf{O}, \mathrm{d}\mathbf{O} \in \mathbb{R}^{T \times d}, \mathbf{I}, \mathbf{A} \in \mathbb{R}^{T \times m}$
    into $N = \left\lceil \frac{T}{C} \right\rceil$ blocks           ▷ $C$ is chunk size
  **function** gsa_bwd($\mathbf{Q}, \mathbf{K}, \mathbf{V}, \mathbf{S}, \overrightarrow{\mathcal{A}}, \mathrm{d}\mathbf{O}$, GATE_K)
    On chip: construct causal mask $\mathbf{M} \in \mathbb{R}^{C \times C}$
    **for** $n \leftarrow N, 1$ **do**           ▷ in reverse order
      Store $\mathrm{d}\mathbf{S}$ in HBM as $\mathrm{d}\mathbf{S}_{[n]}$           ▷ Initialize $\mathrm{d}\mathbf{S} = \mathbf{0}$
      Load $\mathbf{Q}_{[n]}, \overrightarrow{\mathcal{A}}_{[n]}, \mathrm{d}\mathbf{O}_{[n]}$ to SRAM
      On chip:
      **if** GATE_K **then**
        $\mathrm{d}\mathbf{S} \leftarrow \mathrm{Diag}(\overrightarrow{\mathcal{A}}_{[i],C})\mathrm{d}\mathbf{S} + (\mathbf{Q}_{[n]} \odot \overrightarrow{\mathcal{A}}_{[n]})^\top \mathrm{d}\mathbf{O}_{[n]}$
      **else**
        $\mathrm{d}\mathbf{S} \leftarrow \mathrm{d}\mathbf{S}\mathrm{Diag}(\overrightarrow{\mathcal{A}}_{[i],C}) + \mathbf{Q}_{[n]}^\top(\mathrm{d}\mathbf{O}_{[n]} \odot \overrightarrow{\mathcal{A}}_{[n]})$
    **parfor** $n \leftarrow 1, N$ **do**
      Load $\mathbf{Q}_{[n]}, \mathbf{K}_{[n]}, \mathbf{V}_{[n]}, \mathrm{d}\mathbf{O}_{[n]} \in \mathbb{R}^{C \times d}$
        $\mathbf{S}_{[n]}, \mathrm{d}\mathbf{S}_{[n]} \in \mathbb{R}^{d \times d}$ to SRAM
      On chip:           ▷ Recompute $\overleftarrow{\mathcal{A}}_{[n]}, \bar{\mathbf{Q}}_{[n]}, \bar{\mathbf{K}}_{[n]}, \bar{\mathbf{V}}_{[n]}, \mathbf{P}$
      **if** GATE_K **then**
        $\mathrm{d}\mathbf{P} \leftarrow (\mathrm{d}\mathbf{O}_{[n]}\mathbf{V}_{[n]}^\top) \odot \mathbf{M}$
        $\mathrm{d}\mathbf{Q} \leftarrow (\mathrm{d}\mathbf{O}_{[n]}\mathbf{S} + \mathrm{d}\mathbf{P}\bar{\mathbf{K}}_{[n]}^\top) \odot \overrightarrow{\mathcal{A}}_{[n]}$
        $\mathrm{d}\mathbf{K} \leftarrow (\mathbf{V}_{[n]}\mathrm{d}\mathbf{S}^\top + \mathrm{d}\mathbf{P}^\top\bar{\mathbf{Q}}_{[n]}) \odot \overleftarrow{\mathcal{A}}_{[n]}$
        $\mathrm{d}\mathbf{V} \leftarrow \bar{\mathbf{K}}_{[n]}\mathrm{d}\mathbf{S}_{[n]} + \mathbf{P}^\top \mathrm{d}\mathbf{O}_{[n]}$
      **else**
        $\mathrm{d}\mathbf{P} \leftarrow (\mathrm{d}\mathbf{O}_{[n]}\bar{\mathbf{V}}_{[n]}^\top) \odot \mathbf{M}$
        $\mathrm{d}\mathbf{Q} \leftarrow \mathrm{d}\mathbf{O}_{[n]}\mathbf{S}^\top + \mathrm{d}\mathbf{P}\mathbf{K}_{[n]}$
        $\mathrm{d}\mathbf{K} \leftarrow \bar{\mathbf{V}}_{[n]}\mathrm{d}\mathbf{S}^\top + \mathrm{d}\mathbf{P}^\top\mathbf{Q}_{[n]}$
        $\mathrm{d}\mathbf{V} \leftarrow (\mathbf{K}_{[n]}\mathrm{d}\mathbf{S}_{[n]} + \mathbf{P}^\top \mathrm{d}\mathbf{O}_{[n]}) \odot \overleftarrow{\mathcal{A}}_{[n]}$
      Write $\mathrm{d}\mathbf{Q}, \mathrm{d}\mathbf{K}, \mathrm{d}\mathbf{V}$ to HBM as $\mathrm{d}\mathbf{Q}_{[n]}, \mathrm{d}\mathbf{K}_{[n]}, \mathrm{d}\mathbf{V}_{[n]}$
    **return** $\mathrm{d}\mathbf{Q}_{[1,\ldots,N]}, \mathrm{d}\mathbf{K}_{[1,\ldots,N]}, \mathrm{d}\mathbf{V}_{[1,\ldots,N]}$

  Recompute $\overrightarrow{\mathcal{A}}, \mathbf{S}^k, \mathbf{S}^v$
  $\mathrm{d}\mathbf{Q}^v, \mathrm{d}\mathbf{I}^v, \mathrm{d}\mathbf{V} \leftarrow$ gsa_bwd($\mathbf{Q}, \mathbf{I}, \mathbf{V}, \mathbf{S}^v, \overrightarrow{\mathcal{A}}, \mathrm{d}\mathbf{O}$, False)
  $\mathrm{d}\mathbf{O}^k \leftarrow \mathrm{d}$ softmax($\mathbf{O}^k, \mathrm{d}\mathbf{Q}^v$)           ▷ softmax gradients
  $\mathrm{d}\mathbf{Q}, \mathrm{d}\mathbf{K}, \mathrm{d}\mathbf{I}^k \leftarrow$ gsa_bwd($\mathbf{Q}, \mathbf{K}, \mathbf{I}, \mathbf{S}^k, \overrightarrow{\mathcal{A}}, \mathrm{d}\mathbf{O}^k$, True)
  $\mathrm{d}\mathbf{I} \leftarrow \mathrm{d}\mathbf{I}^k + \mathrm{d}\mathbf{I}^v$
  $\mathrm{d}\mathbf{A} \leftarrow$ reversed_cumsum($\mathbf{Q} \odot \mathrm{d}\mathbf{Q} - \mathbf{K} \odot \mathrm{d}\mathbf{K} + \mathbf{O} \odot \mathrm{d}\mathbf{O} - \mathbf{V} \odot \mathrm{d}\mathbf{V}$)
  **return** $\mathrm{d}\mathbf{Q}, \mathrm{d}\mathbf{K}, \mathrm{d}\mathbf{V}, \mathrm{d}\mathbf{I}, \mathrm{d}\mathbf{A}$

In the second pass, the output $\mathbf{O}^k$ from the first pass, after the application of the softmax function, serves as the queries $\mathbf{Q}^v$,

$$\mathbf{Q}^v = \mathrm{softmax}(\ \mathbf{O}^k\ )$$

and $\mathbf{I}/\mathbf{V}$ are used as the key/value vectors, respectively. The final GSA output $\mathbf{O}$ is obtained as follows:

$$\mathbf{O}_{[i]} = \mathbf{Q}_{[i]}^v\ \mathbf{S}_{[i-1]}^v + ((\ \mathbf{Q}_{[i]}^v\ \mathbf{I}_{[i]}^\top) \odot \mathbf{M})\bar{\mathbf{V}}_{[i]} \in \mathbb{R}^{C \times d}$$

Unlike in the first pass, $\overrightarrow{\mathcal{A}}, \overleftarrow{\mathcal{A}}$ is absorbed into $\mathbf{V}, \mathbf{O}$ rather than $\mathbf{Q}, \mathbf{K}$.

During the backward pass, computing the gradients of $\mathbf{Q}, \mathbf{K}, \mathbf{V}, \mathbf{I}, \mathbf{A}$ involves variables already computed in the forward pass. However, directly saving all intermediate results can pose severe challenges for memory management. To address this issue, we adopt gradient checkpointing [11] to trade off memory consumption for recomputation. In addition to the input $\mathbf{Q}, \mathbf{K}, \mathbf{V}, \mathbf{I}, \mathbf{A}$, we selectively save only the output of the first GLA pass, which significantly reduces memory consumption (Figure 4b).

Similar to the forward pass, the backward pass involves two GLA backward passes as well, but in the reverse order. The final gradient $\mathrm{d}\mathbf{I}$ is obtained by combining the gradients from these computations, i.e., $\mathrm{d}\mathbf{I} = \mathrm{d}\mathbf{I}^k + \mathrm{d}\mathbf{I}^v$. The forget gate gradient can be decomposed into two parts: $\mathbf{Q} \odot \mathrm{d}\mathbf{Q} - \mathbf{K} \odot \mathrm{d}\mathbf{K}$ and $\mathbf{O} \odot \mathrm{d}\mathbf{O} - \mathbf{V} \odot \mathrm{d}\mathbf{V}$ (cf. §C in [96]). The reversed cumulative sum in the backward pass corresponds to the cumulative sum computed in the preprocessing step of the forward pass.

We provide a PyTorch implementation for the above algorithm with chunkwise parallelism in Listing 1.

```python
def gsa_fwd_k(q, k, v, g, C):
    '''
    q/k/v:
        query, key, value of shape [NC, C, K|V]
    g:
        local cumulative product of forget gate in log space
    C:
        chunk size
    '''
    # NC: number of chunks
    # K: query/key head dimension
    # V: value head dimension
    NC, C, K, V = *q.shape, v.shape[-1]
    # [K, V]
    s = q.new_zeros(K, V)
    # [NC, C, V]
    o = torch.empty_like(v)

    for i in range(0, NC):
        # [C, K|V] chunking
        c_q, c_k, c_v, c_g = q[i], k[i], v[i], g[i]
        # the last g of each chunk
        c_gn = c_g[-1]
        # inter-chunk w/ matmul
        c_vg, c_gn = c_v * (c_gn - c_g).exp(), c_gn.exp()
        # [C, V]
        c_o_inter = (c_q @ s) * c_g.exp()
        # hidden state update
        s = c_gn * s + c_k.t() @ c_vg

        # intra-chunk
        # [C, C]
        c_A = c_q @ c_k.t()
        # [C, V]
        c_o_intra = torch.zeros_like(c_v)
        for j in range(0, C // 16):
            t = slice(j * 16, j * 16 + 16)
            # [16, K|V] subchunking
            s_A, s_v, s_g = c_A[t], c_v[t], c_g[t]
            s_o = q.new_zeros(16, V)

            # inter-subchunk w/ matmul
            s_gn = s_g[0]
            for si in range(0, j):
                u = slice(si * 16, si * 16 + 16)
                s_o += s_A[:, u] @ (c_v[u] * (s_gn - c_g[u]).exp())
            s_o *= (s_g - s_gn).exp()
            # intra-subchunk w/o matmul
            for si in range(16):
                for sj in range(si + 1):
                    s_o[si] += s_A[si, j * 16 + sj] * s_v[sj] * (s_g[si] - s_g[sj]).exp()
            c_o_intra[t] = s_o
        # [C, V]
        o[i] = c_o_inter + c_o_intra
    return o

def gsa_fwd_v(q, k, v, g, C):
    NC, C, K, V = *q.shape, v.shape[-1]
    s = q.new_zeros(K, V)
    o = torch.empty_like(v)

    for i in range(0, NC):
```

```
64          # [C, K|V] chunking
65          c_q, c_k, c_v, c_g = q[i], k[i], v[i], g[i]
66          # the last g of each chunk
67          c_gn = c_g[-1]
68          # inter-chunk w/ matmul
69          c_qg, c_kg, c_gn = c_q * c_g.exp(), c_k * (c_gn - c_g).exp(), c_gn.exp()
70          # [C, V]
71          c_o_inter = c_qg @ s
72          # hidden state update
73          s = c_gn[:, None] * s + c_kg.t() @ c_v
74
75          # intra-chunk
76          c_A = q.new_zeros(C, C)
77          for j in range(0, C // 16):
78              t = slice(j * 16, j * 16 + 16)
79              # [16, K|V] subchunking
80              s_q, s_k, s_g = c_q[t], c_k[t], c_g[t]
81              s_A = q.new_zeros(16, 16)
82
83              # intra-subchunk w/o matmul
84              for si in range(16):
85                  for sj in range(si + 1):
86                      s_A[si, sj] = torch.sum(s_q[si] * s_k[sj] * (s_g[si] - s_g[sj]).exp())
87              c_A[t, t] = s_A
88              # inter-subchunk w/ matmul
89              s_gn = s_g[0]
90              s_qg = s_q * (s_g - s_gn).exp()
91              for si in range(0, j):
92                  u = slice(si * 16, si * 16 + 16)
93                  c_A[t, u] = s_qg @ (c_k[u] * (s_gn - c_g[u]).exp()).t()
94          c_o_intra = c_A @ c_v
95          # [C, V]
96          o[i] = c_o_inter + c_o_intra
97      return o
98
99
100 def gsa(q, k, v, s, g):
101     T, M = s.shape
102     # reshape each input to [NC, C, K|V]
103     q, k, v, s, g = map(lambda x: x.view(-1, C, x.shape[-1]), (q, k, v, s, g))
104     # local compute of cumulative product of decay
105     # [NC, C, K]
106     g = g.cumsum(1)
107     ok = gsa_fwd_k(q, k, s, g, M)
108     qv = ok.softmax(-1)
109     o = gsa_fwd_v(qv, s, v, g, M)
110     return o.view(T, -1)
```

**Listing 1:** Pseudo PyTorch-style code snippet for GSA with chunkwise parallelism. For brevity, we omit the dimensions of batch size and number of heads. Notably, unlike Algorithm 1, we obtain the intra outputs via a secondary chunking strategy in Line 31-52 and Line 75-94, as utilized by GLA [96], to ensure numerical stability.

## C    Experimental Setup

### C.1    Language Modeling

We compare GSA with the following strong Transformers with modern architectural recipes as well as other recent subquadratic architectures:

- Xfmr++ [86]: Llama-like architectures that enhance the vanilla Transformer by using Rotary position embeddings [80] and GLU [78].

- Mamba [29]: State-space models with *data-dependent* decay.
- RetNet [82]: Linear attention with non-learnable, *data-independent head-wise* decay and rotary embedding.
- GLA [96]: Linear attention with elementwise *data-dependent* decay.
- HGRN2 [69]: Gated Linear RNN with state expansion, or GLA with improved parameterization.

**Setup.** For a fair comparison, all models are trained from scratch with the same training recipes. We utilize a subset of 100B tokens picked from the Slimpajama dataset [79]. The input tokens are processed using the Mistral tokenizer [39] [11]. We use AdamW [50] with a weight decay $0.01$ as the optimizer. During training, the learning rate is first warmed up to $3 \times 10^{-4}$ in the first 1B tokens, and then decayed to $3 \times 10^{-5}$ gradually with a cosine schedule. The number of attention heads is set to 4 and 5 for 1.3B and 2.7B models, respectively. The number of memory slots is uniformly set to 64 for all models. We utilize the open-sourced Triton-based library FLA [95] to run all compared models.

We ran all models on 32 Nvidia H800 GPUs. To facilitate distributed training and accelerate the process, we utilized the DeepSpeed framework and fused all necessary modules, including ROPE, cross-entropy, and LayerNorm, following the practice of [102]. The training of a GSA model with 2.7B parameters took approximately 2 days, while the 1.3B model required 1 day to complete training.

**Remark on state size.** Let the model dimension be denoted as $d$. Mamba expands the value projection to $2d$ and uses a state expansion ratio of 16, resulting in a state size of $32d$ per layer. Since Mamba also replaces the FFN with a Mamba layer, this effectively doubles both the number of recurrent layers and the state size, leading to a total recurrent state size of $64Ld$.

Similarly, RetNet expands the value projection to $2d$ and sets the head dimension of queries/keys to be half that of the value head dimension. RetNet also reduces the number of heads to increase the head dimensions of queries and keys. We fix the query/key head dimension to 256 and adjust the number of heads accordingly, resulting in a recurrent state size of $512d$ per layer and $512Ld$ in total.

GLA does not expand the value projection but reduces the head dimensions of queries and keys to half of the value head dimension to save parameters for the Swish output gate, ensuring each layer contains $4d^2$ parameters. We fix the query/key head dimension to 256 and adjust the number of heads accordingly, resulting in a recurrent state size of $256d$ per layer and $256Ld$ in total.

HGRN2 follows a similar approach to GLA but without the Swish output gate, keeping the head dimensions of queries/keys and values equal, as in standard softmax attention, while still retaining $4d^2$ total parameters per recurrent layer. We set the head dimension to 128, resulting in a recurrent state size of $128d$ per layer and $128Ld$ in total.

GSA maintains hidden states for both keys and values, so each layer contains a recurrent state size of $2 \times 64 \times d$. We fix the state expansion (i.e., number of slots) to $64^{12}$, resulting in a total recurrent state size of $128Ld$.

---

[11] https://huggingface.co/mistralai/Mistral-7B-v0.1
[12] Note that in this case, the number of heads is independent of the state expansion ratio

