# OpenReview forum: "Gated Slot Attention for Efficient Linear-Time Sequence Modeling"
_NeurIPS.cc/2024/Conference — NeurIPS 2024 poster_

### Official Review · Reviewer_eYuC · 2024-06-28

**Soundness:** 3
**Presentation:** 3
**Contribution:** 3
**Rating:** 6
**Confidence:** 3

**Summary:**

The paper introduces Gated Slot Attention (GSA), an enhancement of Gated Linear Attention (GLA), aimed at improving the efficiency of sequence modeling. GSA incorporates a selective gating mechanism to manage memory updates, leveraging a two-pass GLA structure. This approach allows GSA to be more context-aware and to retain a bounded memory footprint, making it suitable for long sequence tasks.
The key improvement is the incorporation of a softmax operation to retain sharp attention distributions, which enhances sequence modeling by reducing dilution.

**Strengths:**

- The GSA's gated update mechanism and two-pass structure offer a straightforward yet effective enhancement over GLA. This design ensures that memory usage remains bounded and manageable, which is critical for handling long sequences efficiently.
- The experimental results provided by the authors demonstrate the advantages of the GSA mechanism. The results highlight GSA's ability to improve performance in sequence modeling tasks, validating the practical effectiveness of the proposed approach.

**Weaknesses:**

- While GSA presents improvements, it largely builds on existing GLA techniques. The enhancements, though valuable, might be seen as incremental rather than revolutionary, potentially limiting the perceived impact of the work.
- Despite efforts to reduce computational overhead, the softmax operation on $QK^T$ still retains a quadratic complexity in training. This raises questions about the authors' claim of linear-time sequence modeling in the title, which could be misleading.
- The authors do not specify explicitly whether GSA inherits the recurrent architecture of GLA. If this is the case, it would be necessary to compare GSA to other recurrent models like RWKV to provide a comprehensive evaluation of its performance and advantages in sequence modeling tasks.

**Questions:**

How does the inference performance of GSA compare to other SOTA models in terms of speed and memory usage?

**Limitations:**

Yes

---

> ### Author Rebuttal · Authors · 2024-08-06
>
> **Q:** *While GSA presents improvements, it largely builds on existing GLA techniques. The
> enhancements, though valuable, might be seen as incremental rather than revolutionary, potentially limiting the perceived impact of the work.*
>
> **A:** We appreciate your feedback and respectfully argue that GSA represents substantial advancements over existing techniques:
>
> * GSA is motivated by viewing $\mathbf{K}$ and $\mathbf{V}$ as memories and linearizing standard attention by managing $\mathbf{K}$ and $\mathbf{V}$ into constant number of memory slots. This perspective drives our 2-pass recurrence formulation: $\mathrm{softmax}(\mathbf{Q}\tilde{\mathbf{K}}^\top)\tilde{\mathbf{V}}$.
> This significantly differ from exising linear attention that conduct linearization by rearranging the computation order of $\mathbf{Q}\mathbf{K}^\top\mathbf{V}$.
> * The 2-pass recurrence with context-aware queries demonstrates significant improvements in retrieval tasks (FDA, SQUAD, SWDE), showcasing the practical impact of our approach.
> * GSA maintains the general form of standard attention while offering $\mathbf{K}\mathbf{V}$ linearization. The $\mathrm{softmax}$ spikiness [1] notably enhances continual-pretraining results, paving the way for efficient scaling to larger models.
>
> ---
>
> **Q:** *Despite efforts to reduce computational overhead, the softmax operation on $\mathbf{Q}\mathbf{K}^\top$ still retains a quadratic complexity in training. This raises questions about the authors' claim of linear-time sequence modeling in the title, which could be misleading.*
>
> **A:** Thank you for your concern! We would like to clarify that in GSA, the $\mathrm{softmax}$ is applied over $\mathbf{Q}\tilde{\mathbf{K}}^\top$ rather than $\mathbf{Q}\mathbf{K}^\top$ in standard attention (SA), as presented in Section 2.2.
> Unlike $\mathbf{K}\in \mathbb{R}^{T\times d}$, $\tilde{\mathbf{K}}\in \mathbb{R}^{m\times d}$ is derived by passing $\mathbf{K}$ along with $\mathbf{\phi}\in \mathbb{R}^{T\times m}$ through GLA recurrence, compressing it to a constant size of $m\times d$.
> This crucial difference enables GSA to reduce the training complexity from quadratic to linear time, supporting our claim of linear-time sequence modeling.
>
> We will ensure this distinction is more explicitly stated in the revised version. Thank you.
>
> ---
>
> **Q:** *The authors do not specify explicitly whether GSA inherits the recurrent architecture of GLA. If this is the case, it would be necessary to compare GSA to other recurrent models like RWKV to provide a comprehensive evaluation of its performance and advantages in sequence modeling tasks.*
>
> **A:** We would like to clarify that: as detailed in Section 3, GSA's output is indeed derived from $\tilde{\mathbf{K}}_t^\top \mathbf{q}_t \in \mathbb{R}^d$, where $\tilde{\mathbf{K}}$ and $\tilde{\mathbf{V}}$ maintain constant sizes regardless of sequence length. Both $\tilde{\mathbf{K}}$ and $\tilde{\mathbf{V}}$ involves one GLA recurrence.
> We provide hardware-efficient implementations for the 2-pass operations to enhance training efficiency. Full algorithm details are available in Appendix A.
>
> Regarding comparisons with other recurrent models, we include comparisons with RWKV6 in Table 5 in the paper.
> Our results show that GSA significantly outperforms RWKV6 under controlled settings, while RWKV6 is trained on billions of tokens and GSA uses continual pretraining.
> It's worth noting that while RWKV6 shares similarities with GLA in terms of data-dependent gating and matrix-formed hidden states, it employs additional short convolutions at the expense of efficiency.
>
> Due to limited computational resources, we haven't provided results for RWKV6 models trained from scratch, nor have we employed the specific techniques used by RWKV6, to ensure fair comparisons.
> We commit to improving the clarity of our presentation in the next version of our paper.
>
> ---
>
> **Q:** *How does the inference performance of GSA compare to other SOTA models in terms of speed and memory usage?*
>
> **A:** During inference, GSA demonstrates similar theoretical inference performance compared to other SOTA models :
> * GSA needs $O(1)$ space for hidden states, similar to GLA and other linear models, while SA requires $O(N)$ KV cache.
> * GSA requires $O(1)$ time per step to access memory and generate outputs as in GLA.
>
> Our empirical speed evaluations for 7B models with 2k prefix support these advantages:
> * GSA and GLA consume similar 15GiB memory, while SA requires an additional 3GiB due to KV cache requirements for further generation.
> * Regarding speed, the latency for GSA is 13s, comparable to GLA (13.4s) and SA (12s), aligning with our findings in Tables 2 & 3 of our paper.
>
> It's important to note that we have not yet fully optimize the recurrent kernel for speeding up the generation stage, nor reducing the I/O overhead through customization.
> We are committed to exploring these optimizations in future work to fully realize GSA's theoretical inference advantages and enhance its real-world performance.
> Thank you.
>
> [1] The Hedgehog & the Porcupine: Expressive Linear Attentions with Softmax Mimicry: https://arxiv.org/abs/2402.04347

---

> > ### Author Response · Authors · 2024-08-12
> >
> > Dear Reviewer eYuC,
> >
> > Could you please let us know if our response has addressed your concerns? If you have any further questions, please feel free to raise them at any time.

---

> ### Author Response · Authors · 2024-08-13
> **Response to Reviewer eYuC: additional inference efficiency comparisons**
>
> To provide more clarity, we conducted additional comparisons on an H800 GPU, optimizing our inference kernel for GSA to perform one-by-one generation.
> This optimization significantly reduces I/O overhead for 2-pass recurrences, which is crucial for auto-regressive generation.
> Our updated results show:
>
> |     | Transformer++ |          GLA |          GSA |
> | --- | ------------: | -----------: | -----------: |
> | 2k  |   85.0 (14.9) |  77.3 (14.0) |  78.0 (14.1) |
> | 4k  |  169.4 (15.5) | 140.9 (13.9) | 141.7 (14.1) |
> | 8k  |  350.5 (16.6) | 274.6 (13.9) | 269.2 (14.0) |
> | 16k |  783.1 (18.8) | 528.0 (13.9) | 517.1 (14.1) |
>
>
> We compare the inference latency (seconds) as well as the memory consumption of Transformer++, GLA, and GSA for a single sequence.
> By varying the generation length from 2k to 16k:
> * GSA maintains consistent memory usage across different sequence lengths, unlike Transformer++ which consumes up to 4GiB more for 16k sequences.
> * GSA shows comparable or slightly better inference speed than GLA, especially for longer sequences, thanks to our optimized fused kernel.
>
> These findings underscore GSA's competitive performance in real-world scenarios, combining the theoretical advantages of linear attention models with practical optimizations.
> We are committed to further exploring and implementing optimizations to fully realize GSA's potential and enhance its real-world performance.
>
> We hope this clarifies our contribution and strengthens our paper's position.

---

### Official Review · Reviewer_LU5r · 2024-07-08

**Soundness:** 3
**Presentation:** 3
**Contribution:** 2
**Rating:** 6
**Confidence:** 4

**Summary:**

A major challenge is storing information in a bounded number of memory slots. This work builds on ideas from ABC and gated linear attention. ABC recursively updates the bounded-memory keys and values states over time, and computes a softmax with the queries at timestep t to produce the output at t. Gated linear attention and Mamba use data-dependent decays over time to better decide what information to keep versus throw away given the limited memory. GLA, unlike Mamba, enables the use of GPU tensor cores through its parameterization.

GSA retains the selective gating from GLA and the update rules from GSA. The architecture remains chunk-wise parallelizable (like GLA) and provides inductive biases that address limitations of GLA and ABC respectively. GSA is validated up to 2.7Bn parameters from scratch and 7Bn parameters in continual training, providing promising results on standard LM-eval harness benchmark tasks.

**Strengths:**

The writing and contextualization of the contributions with respect to prior work are very clear.

The architectural inductive baises from GSA versus prior linear recurrent architectures are compelling and very interesting. For e.g., “In GLA the query vector is only dependent on the current input token, while in GSA, the query vector is the output of the first GLA pass and thereby is aware of the entire historical information and in this sense more context aware”. Two passes over the input could help the model make better decisions about what information to keep versus throw away given the limited recurrent memory.

GSA is trained from scratch to relatively large scales (2.7Bn parameters, 100B tokens) and demonstrates high quality on LM-eval harness tasks compared to the baselines, despite using less memory than GLA.

GSA can be implemented efficiently by adopting off-the-shelf algorithms from flash linear attention.

**Weaknesses:**

Building on my comment on the compelling properties of GSA, the paper does not extensively show how these properties make a difference in language modeling. It would be interesting to know how the “context aware query vectors” help GSA on real-world data by comparing to models without this property. OR for the authors to include discussion on what kind of sequences this property might help with. The architectural modifications are not tied to empirical insights beyond lm-eval harness scores in the current submission.

GSA is only evaluated on LM-eval harness. It is known that models perform somewhat similarly on these short-context tasks with very little memory (Arora et al, 2024) and mentioned in the paper’s limitations section. The paper does not dive into any tasks that require longer context reasoning or retrieval, making it unclear how GSA behaves.
- "GSA only needs half the recurrent state size of GLA and a quarter the recurrent size of RetNet, while having better performance, thus enjoying a lower memory footprint during inference". The claim is not fully validated unless the GSA models are evaluated on tasks that stress-test memory utilization (like retrieval, long-context tasks).

The MMLU scores of the continually fine-tuned 7B checkpoints remains very low, just like the baseline – the other scores are roughly comparable across models. There is no analysis as to why this is. The paper claims that GSA is drastically better than SUPRA, but the delta is small (<1 point on average), so it is not clear what is meant by this claim.

**Questions:**

1. Why do the authors believe the current set of benchmarks is sufficient? Is it possible to include results on retrieval and longer context benchmarks?

2. Can the authors perform error analysis on the continually trained models to help understand why SUPRA and GSA perform poorly on MMLU?

3. Can the authors provide more concrete hypotheses as to where context-aware query tokens (resulting from the first GLA pass) could help in real language modeling settings?

In the introduction, “slots” are not defined. They are first defined in section 2.

**Limitations:**

The authors do mention that they ignore retrieval style tasks in their analysis (in Section 6), however if the paper makes claims about memory-efficiency then it is important to evaluate quality on this style of tasks. This is because it is known that there are fundamental memory and quality tradeoffs.

If the reviewers address these concerns, I will consider increasing my score!

---

> ### Author Rebuttal · Authors · 2024-08-06
>
> Thank you for your thoughtful feedback and insightful questions.
> We promise to revise the paper to address your concerns and make the presentation more clear and comprehensive in the next version.
>
>
> ---
>
> **Q:** *It would be interesting to know how the "context aware query vectors" help GSA on real-world data by comparing to models without this property.
>  Why do the authors believe the current set of benchmarks is sufficient? Is it possible to include results on retrieval and longer context benchmarks?*
>
> **A:** Thank you for your very constructive suggestions.
> In the above table, we have additionally included results to validate the effectiveness of "context aware query vectors":
> * We conduct experiments on three tasks proposed by Arora et al, 2024 [1]: FDA, SQUAD, and SWDE, showing **significant improvements in GSA's performance compared to Mamba, RetNet, and GLA on these recall-intensive tasks**. Despite smaller memory capacity, GSA outperforms Mamba, RetNet, and GLA by an average of 5.9, 7.7, and 4.8 points, respectively.
> * We report results on long context benchmarks reported by Xiong et al. 2023 [2]. It is clear that **GSA can be well extrapolated to 16k give 2k training context length**. GSA exhibits better extrapolation than GLA and RetNet on Qasper, NarrativeQA, QuALITY and QMSum.
>
> These results validate GSA's strong performance on information extraction tasks, which we attribute to more effective memory utilization enabled by "context aware query vectors."
> We are planning to inspect the performance of GSA scaled to larger model size and number of slots.
>
> ---
> **Q:** *The MMLU scores of the continually fine-tuned 7B checkpoints remains very low, just like the baseline – the other scores are roughly comparable across models.
> The paper claims that GSA is drastically better than SUPRA, but the delta is small (<1 point on average), so it is not clear what is meant by this claim.
> Can the authors perform error analysis on the continually trained models to help understand why SUPRA and GSA perform poorly on MMLU?*
>
> **A:** After carefully diving into the details of SUPRA, we updated the continual pretraining results of GSA models trained with low learning rate of $3×10^{-5}$ combined with slow decay scheduler.
> From the above table, we observe that GSA significantly outperforms RetNet and GLA across various tasks, including NQ, TriviaQA, and BBH.
> Notably, for challenging reasoning / language understanding tasks of BBH and MMLU, GSA outperforms GLA by 2.7 and 4.0 points, indicating that GSA's formulation, which is similar to standard attention (SA), allows it to retain more capabilities when finetuned from another SA LLM.
>
> In the bottom lines we further provide the results of continual pretraining 100B tokens: GSA greatly surpasses SUPRA by 2.2, 5.4, 8.7 and 6.0 points, respectively.
>
> [1] Simple linear attention language models balance the recall-throughput tradeoff: https://arxiv.org/abs/2402.18668
>
> [2] Effective Long-Context Scaling of Foundation Models: https://arxiv.org/abs/2309.16039

---

> > ### Author Response · Authors · 2024-08-12
> >
> > Dear Reviewer LU5r,
> >
> > Could you please let us know if our response has addressed your concerns? If you have any further questions, please feel free to raise them at any time.

---

> > > ### Comment · Reviewer_LU5r · 2024-08-12
> > > **Thank you for your response**
> > >
> > > The new experiments are helpful.
> > >
> > > Can the authors further explain why GSA should be faster or equal in speed to GLA, despite using two-passes? The writing around this point could be clarified.

---

> ### Author Response · Authors · 2024-08-13
> **Response to Reviewer LU5r13: speed analysis**
>
> Thank you for your question and we are happy to provide further clarifications on the speed comparisons during both training and inference:
>
> **Training**
>
> For sequence length $N$, chunk $C$, and head dimension $d$:
> * GLA with chunkwise parallelism requires $O(NC d + N d^2)$ FLOPs and $O(Nd)$ additional memory for forget gates [1].
> * GSA with $m$ memory slots (modeled as two-pass GLA recurrences) requires $O(NC m + NC d + 2N md)$ FLOPs and $O(Nm+Nd)$ memory.
>
> In our experiments, we set $m=64$ to leverage tensor core accelerations, as matmuls operating on $64\times 64$ tiles are shown to be highly hardware-efficient [2].
> For models > 1B, $d$ is larger than 512.
> In these cases, despite the two-pass nature, GSA's complexity ($O(NC m + NC d + 2N md)$) can be lower than that of GLA ($O(NC d + N d^2)$) since $2md=128d < d^2$.
>
> **Inference**
>
> Both GLA and GSA maintain constant time and memory complexity during token-by-token generation. We also implemented fused kernels to reduce the IO overhead for operations like $\mathtt{softmax}$, potentially making GSA faster than GLA implementations in FLA [3].
>
> For comparison, Transformers require quadratic time complexity for training and $O(N)$ time and memory for inference.
>
> To illustrate the efficiency, we provide inference latency (seconds) for a single sentence beyond the training speed / memory analysis in Table 2 of our paper:
>
> |     | Transformer++ |   GLA |   GSA |
> | --- | ------------: | ----: | ----: |
> | 2k  |          85.0 |  77.3 |  78.0 |
> | 4k  |         169.4 | 140.9 | 141.7 |
> | 8k  |         350.5 | 274.6 | 269.2 |
> | 16k |         783.1 | 528.0 | 517.1 |
>
> Note: We used the basic Huggingface `model.generate` API for demonstration, leaving room for further optimization.
> * Transformer++ underperforms even at moderate lengths (2k) and scales poorly.
> * GSA performs comparably to GLA, slightly outperforming it for sequences >8k due to our optimized fused inference implementations.
>
> We will include a detailed speed analysis for both training and inference in our revised manuscript.
> Thank you for bringing this to our attention.
>
> [1] Gated Linear Attention Transformers with Hardware-Efficient Training: https://arxiv.org/abc/2312.06635
>
> [2] GPUs Go Brrr: https://hazyresearch.stanford.edu/blog/2024-05-12-tk
>
> [3] FLA: A Triton-Based Library for Hardware-Efficient Implementations of Linear Attention Mechanism: https://github.com/sustcsonglin/flash-linear-attention

---

> > ### Comment · Reviewer_LU5r · 2024-08-13
> > **Thanks**
> >
> > I have raised my score

---

> > > ### Author Response · Authors · 2024-08-13
> > >
> > > Thank you very much! We are delighted that our response has addressed your concerns. We will incorporate these new results in the next iteration of the paper.

---

### Official Review · Reviewer_AbD2 · 2024-07-13

**Soundness:** 3
**Presentation:** 2
**Contribution:** 2
**Rating:** 5
**Confidence:** 3

**Summary:**

The paper explores a new variant of attention with bounded memory to reduce the growing memory size and thus mitigates the memory-intensive challenges of Transformers. The key idea is to set a memory bound, with a predetermined number of usable memory slots, and a gating mechanism to select or mix KV vectors from the previous step.

**Strengths:**

The paper proposes a variant of attention mechanism with bounded memory, which seems to be an original contribution.

**Weaknesses:**

* The paper could clarify the significance of its proposed method. The results show not so much improvements compared with prior work in terms of performance and memory costs. The effectiveness and efficiency of the proposed method remain unclear.
* The writing on the presentation of the proposed method and the evaluation and insightful discussion could be improved.

**Questions:**

* Fig. 1 can be better illustrated and explained how the proposed Gate Slot Attention works.
* What is the reason for choosing memory slots to fit in SRAMs? Particularly when the claim is for efficient training.
* What is the contribution when the memory footprint as shown in Table 2 is larger than other methods?
* What are the overheads, say additional parameters and operations, as introduced by this work?
* Some key results could be mentioned in the abstract and introduction.
* Fix the extra line-breaker as in line 255?
* The quotation marks used in line 128 to 135 are bizarre.

**Limitations:**

The limitation mentioned in the paper is moderate.

---

> ### Author Rebuttal · Authors · 2024-08-06
>
> We sincerely appreciate your thorough review and insightful feedback.
> We will revise the paper accordingly in the next version to enhance clarity in the abstract, introduction, and discussion sections, polish the writing, fix potential errors, and improve the figures.
>
> ---
>
> **Q:** *The paper could clarify the significance of its proposed method. The results show not so much improvements compared with prior work in terms of performance and memory costs. The effectiveness and efficiency of the proposed method remain unclear.
> What is the contribution when the memory footprint as shown in Table 2 is larger than other methods?*
>
> **A:** We highlight that Gated Slot Attention (GSA) offers significant advantages over existing methods, as shown in the above tables:
> * GSA shows **improved performance on recall-focused tasks** given limited memory capacity compared with Mamba, RetNet and GLA. Especially on SQUAD and SWDE, GSA outperforms other models by 5.4 and 10.8, respectively.
> * GSA's formulation, which mimics the spikiness in standard attention, facilitates effective continual pretraining from well-trained LLMs like Mistral 7B, demonstrating **superior performance on challenging tasks like NQ, TriviaQA, BBH, and MMLU**.
>
> Despite a slight increase in FLOPs, GSA achieves comparable training throughput to GLA (42K vs. 42.5K tokens/s under 8K context length) and outperforms Flash Attention (36.9K tokens/s) greatly.
> Regarding memory footprint, the increase is minimal compared to GLA for sufficiently large head dimensions (d). The benefits in performance and versatility outweigh this small trade-off.
>
> We will provide more detailed discussions on complexity and additional results in the next version of our paper to better highlight these contributions.
>
> ---
>
> **Q:** *What is the reason for choosing memory slots to fit in SRAMs? Particularly when the claim is for efficient training.*
>
> **A:** GSA involves two passes of gated recurrent processes, necessitating hardware-efficient implementations to match GLA's efficiency.
> To address this, we follow the approach in [1], accelerating GSA by fully utilizing tensor cores, which is 16× faster than non-matmul counterparts.
>
> [2] reveals that latencies for tensor-core matmuls of 16×16 remain similar to 64×64 (Figure 1), despite operating on 16×16 tiles.
> Our preliminary experiments confirm these findings, motivating us to choose a number of slots that can be processed efficiently in one matmul pass.
>
> | Slots | PPL (↓) | tokens/s |
> | ----- | ------- | -------- |
> | 32    | 13.74   | 46.7K    |
> | 64    | 13.51   | 44.1K    |
> | 128   | 13.46   | 37.1K    |
>
> As shown in the table, we observe a significant throughput degradation when increasing slots from 64 to 128, while there's a substantial perplexity gap between 32 and 64 slots.
> Consequently, we adopt 64 slots as the default setting, balancing efficiency and performance.
>
> This hardware-efficient implementation allows GSA to match GLA's efficiency while providing improved performance.
> Detailed implementations are provided in Appendix A.
> We will include more discussions and comparisons in the next version of our paper.
> Thank you.
>
> ---
>
> **Q:** *What are the overheads, say additional parameters and operations, as introduced by this work?*
>
> **A:** We appreciate the reviewer's question regarding the overheads introduced by our work.
> We have included some discussions in Section 3.1 of our paper, but we're happy to provide further clarification here.
> * **Parameter allocation**: For 1.3B parameter models, GSA introduces approximately $dhm \approx 0.125 d^2$ additional parameters for forget gate mappings compared to Llama.
> This results in a parameter allocation similar to that of GLA.
> * **Computation Complexity**: The computation complexity of GSA and GLA operation is $O(NCd+Nd^2)$ [3] and $O(NCm+NCd+2Ndm)$, respectively, where N is sequence length, C is chunk size, d is head dimension, and m is number of memory slots.
> While we set $m=64$ and $m\ll d$, the increased complexities are negligible.
>
> Our hardware-efficient implementations demonstrate that GSA performs comparably to GLA in terms of efficiency, as illustrated in Figure 3 in the paper.
> Importantly, this modest increase in parameters and comparable computational efficiency come with significant benefits, which we've discussed in detail in our response to Q1.
> These advantages justify the minimal overhead introduced by our approach.
>
> [1] FLA: A Triton-Based Library for Hardware-Efficient Implementations of Linear Attention Mechanism: https://github.com/sustcsonglin/flash-linear-attention
>
> [2] Simple linear attention language models balance the recall-throughput tradeoff: https://arxiv.org/abs/2402.18668
>
> [3] Gated Linear Attention Transformers with Hardware-Efficient Training: https://arxiv.org/abs/2312.06635

---

> > ### Author Response · Authors · 2024-08-12
> >
> > Dear Reviewer AbD2,
> >
> > Could you please let us know if our response has addressed your concerns? If you have any further questions, please feel free to raise them at any time.

---

> > > ### Author Response · Authors · 2024-08-13
> > >
> > > Dear Reviewer AbD2,
> > >
> > > This is a kind reminder that today is the last day of the author-reviewer discussion period. If you have any concerns, please let us know as soon as possible so that we can address them.

---

> > > > ### Comment · Reviewer_AbD2 · 2024-08-13
> > > >
> > > > Thank you for the clarification, and the new results are helpful. I will revise my evaluation.

---

> ### Author Response · Authors · 2024-08-13
>
> Thank you! We're glad our clarifications were helpful. We'll include these new results in the next version.

---

### Author Rebuttal · Authors · 2024-08-06

We sincerely thank all reviewers for their thorough examination of our work and their insightful feedback.
Your thoughtful comments and questions have significantly enhanced our submission and have been addressed in detail in individual responses.
We have additionally run many new empirical results and analysis, including:
* **Time and memory analysis**: We have added comprehensive efficiency metrics and updated the speed comparisons for different numbers of slots to address Reviewer AbD2's questions regarding computational efficiency.
* **Long-context evaluation**: In response to Reviewer LU5r's concerns, we have performed evaluations on real-world language tasks with extended context, including Qasper, NarrativeQA, QuALITY, and QMSum.
* **Recall-intensive and challenging language reasoning tasks**: To address questions from Reviewers AbD2 and LU5r, we have included results from recall-intensive tasks and more challenging tasks focused on language reasoning to demonstrate the advantages of our GSA approach.

In this shared response, we provide a detailed elaboration of these new results:

### **Recall-intensive tasks**

We evaluated our model on recall-intensive tasks proposed by Arora et al, 2024 [1]:

|            | FDA      | SQUAD    | SWDE     |
| ---------- | -------- | -------- | -------- |
| Mamba      | 9.7      | 33.3     | 34.6     |
| RetNet     | 8.9      | 33.1     | 30.4     |
| GLA        | **11.8** | 25.8     | 43.3     |
| GSA (ours) | 11.3     | **38.7** | **45.4** |

GSA's strong performance, particularly on SQUAD and SWDE, demonstrates that our 2-pass recurrence design enables efficient memory utilization despite limited capacity.

[1] Simple linear attention language models balance the recall-throughput tradeoff: https://arxiv.org/abs/2402.18668

### **Results on challenging language understanding and reasoning tasks**

We expanded our evaluation to include more demanding tasks, following the settings of Llama2 [2] and Mistral:

| Model      | Tokens | NQ       | TriviaQA | BBH      | MMLU     | Avg.     |
| ---------- | ------ | -------- | -------- | -------- | -------- | -------- |
| Mamba      | 1.2T   | 25.4     | 66.2     | 21.5     | 33.2     | 36.5     |
| RWKV6      | 1.4T   | 20.9     | 59.5     | 23.4     | 43.9     | 36.9     |
| SUPRA      | +20B   | -        | -        | -        | 28.0     | -        |
| RetNet     | +20B   | 16.2     | 43.0     | 8.7      | 26.1     | 23.5     |
| GLA        | +20B   | 22.2     | 57.8     | 20.8     | 28.4     | 32.3     |
| GSA (ours) | +20B   | 23.4     | 60.7     | 23.5     | 32.4     | 35.0     |
| SUPRA      | +100B  | 24.7     | 60.4     | 19.8     | 34.1     | 34.7     |
| GSA (ours) | +100B  | **26.9** | **65.8** | **29.3** | **38.1** | **40.0** |

GSA consistently achieves better scores across all tasks compared to SUPRA, RetNet, and GLA at similar token counts and performs competitive with trillion-token models, highlighting its superior in-context learning capabilities and validating our claim that GSA preserves more abilities compared to other linear attention methods under continual-pretraining settings.

[1] Llama 2: Open Foundation and Fine-Tuned Chat Models: https://arxiv.org/abs/2307.09288

### **Length Extrapolation**

In order to showcase the abilities of extrapolating to longer sequences, we conduct additional experiments on long-context tasks, following the settings used in the Llama2-Long report [1].
We tested GSA on four real-world language tasks: Qasper, NarrativeQA, QuALITY, and QMSum.
The input for each task was truncated at 16k tokens following [1], which is 8x longer than our training context length of 2k tokens.
The results are presented in the following table:

|        | Qasper   | NarrativeQA | QuALITY  | QMSum    |
| ------ | -------- | ----------- | -------- | -------- |
| Mamba  | 5.6      | **22.2**    | 27.5     | 0.7      |
| RetNet | 11.1     | 0.0         | 26.2     | 0.0      |
| GLA    | 18.4     | 17.2        | 30.9     | 8.9      |
| GSA    | **18.8** | 19.2        | **32.0** | **10.0** |

These results demonstrate that our GSA model generalizes well to sequences of 16k tokens, despite being trained on much shorter contexts.
Notably, GSA outperforms other linear models across all four tasks, showcasing its robustness and effectiveness in handling long-range dependencies.

[1] Effective Long-Context Scaling of Foundation Models: https://arxiv.org/abs/2309.16039

---

### Author Response · Authors · 2024-08-14
**Response to All Reviewers**

We sincerely thank each reviewer for their thoughtful feedback and recognition of the improvements presented in our rebuttal.
We've added new results on recall-intensive tasks and challenging language understanding and reasoning tasks, as well as length extrapolation to address the concerns raised by Reviewers AbD2 and LU5r. Further analysis on time and memory complexity and comparisons of inference performance were also provided in response to common concerns.

We are pleased that our additional empirical results, detailed analysis, and explanations have addressed your concerns and contributed to raising your scores.
All reviewers have acknowledged our contributions and given positive feedback to our work.

We are committed to further enhancing our work.
In the next version, we will incorporate all suggested revisions to improve clarity, polish the writing, and rectify any remaining errors.
Additionally, we will:
* Release our code: Make our codebase publicly available to facilitate further research and validation of our methods, including Triton kernels for training and inference.
* Provide pretrained weights: Release all pretrained weights, including 1.3B, 2.7B, and 7B parameters.

We sincerely value your input and look forward to sharing our work more widely.
Thank you once again for your detailed reviews and assistance in enhancing our submission.

---

### Decision · Program_Chairs · 2024-09-25

**Decision:**

Accept (poster)

**Comment:**

This paper introduces Gated Slot Attention (GSA) for linear-time sequence modeling, which effectively addresses inefficiencies in traditional attention mechanisms for large language models. All reviewers agree that the method is both novel and impactful. During the rebuttal period, the authors provided additional empirical evidence and clarifications, particularly around memory management and computational performance, which directly addressed the reviewers' concerns. The discussions further solidified that the GSA approach offers substantial advancements over existing methods, validating its effectiveness in a variety of tasks. Based on the strong consensus among reviewers, the thoroughness of the rebuttal, and the high quality of the work, I recommend that the paper be accepted for publication.